# Development of a Bamboo Toothbrush Handle Machine with a Human–Machine Interactive Interface for Optimizing Process Conditions

Bo-Jyun Wang [1], Chia-Hong Lin [2], Wen-Chih Lee [3] and Chun-Ching Hsiao [1,4,*]

[1] Department of Mechanical Design Engineering, National Formosa University, No. 64, Wunhua Rd., Huwei Township 632, Yunlin County, Taiwan; zxc11221012@gmail.com
[2] Yuan-Peing Co., Ltd., No. 16, Heyu 1st Ln., Zhushan Township 557, Nantou County, Taiwan; yuantaiweb@gmail.com
[3] Kuo-An Machinery Co., Ltd., No. 173, Daming Rd., Shengang Dist., Tai-Chung City 429, Taiwan; kuoan@kuoan.com.tw
[4] Smart Machine and Intelligent Manufacturing Research Center, National Formosa University, No. 64, Wunhua Rd., Huwei Township 632, Yunlin County, Taiwan
* Correspondence: cchsiao@nfu.edu.tw; Tel.: +886-5-6315-557

**Abstract:** Non-renewable materials like plastics are widely applied on toothbrush handles and bristles. Polypropylene (PP) or polyethylene (PE) is often used to fabricate the toothbrush handle, while nylon (PA) is used to form the bristle. These plastics are sourced from non-renewable fossil fuels. The primary greenhouse gases—nitrous oxide ($N_2O$) and carbon dioxide ($CO_2$)—in the Earth's atmosphere are released during the production of these plastics. Bamboo can generate 30% more oxygen than most plants and trees, which absorbs twice as much carbon dioxide as trees. A comparison of the cradle-to-grave material requirements between bamboo and plastic toothbrushes reveals that bamboo toothbrushes entail hidden environmental costs. Nevertheless, bamboo toothbrushes can be completely decomposed in the environment, which makes them eco-friendly and sustainable green products. This research aims to develop a bamboo toothbrush handle machine with a human–machine interactive interface and production management for optimizing process conditions. The machine is designed as a double-group to stably mass-produce high-quality bamboo toothbrush handles under optimal process conditions. Although bamboo is a sustainable green material, the shaping process is difficult due to an extremely anisotropic property in the bamboo structures. An improper process condition will induce a rough or scorched surface, which may further cause a tearing crack. The bamboo toothbrush handle milling machine is usually designed by a profiler, which uses various molds to change the shapes and sizes of bamboo toothbrush handles. This machine cannot probe the accurate cutting force for optimizing the cutting operations, paths, and parameters. The proposed equipment with a double-group design includes two storage racks of raw materials, two feeding devices, two exchange clamping devices, and a dual-spindle milling system required to form the shaping process of bamboo toothbrush handles. The whole system is propelled by a computer numerical control (CNC) SYNTEC controller, which can fabricate the bamboo toothbrush handle with various shapes and dimensions. This controller is integrated with a LabVIEW human–machine interactive interface via a Modbus RTU communication protocol. The optimal milling paths, manufacturing methods, and feeding rates are verified by a surveillance system to detect the instant currents of both spindles via the trial-and-error method and mass production. The maximum output of the equipment can reach four bamboo toothbrush handles per minute and 1600 bamboo toothbrush handles per day.

**Keywords:** plastic; toothbrush; bamboo; computer numerical control; damage

## 1. Introduction

Environmental sustainability is a heated discussion in global public health. The world's population faces challenges, including climate change, biodiversity decline, air and water pollution, and ozone depletion. Therefore, healthcare professionals should consider environmental sustainability when recommending medical devices to patients. Toothbrushes are a widely recommended healthcare device. They are inseparable daily necessities and are one of the most important tools for oral cleaning and care. Conventional toothbrushes are extensively made of non-renewable materials like plastic and nylon polymers. Polypropylene (PP) and polyethylene (PE) plastics make up most of the toothbrush handle, while nylon polymers form the bristle. The produced polyethylene releases approximately 1.8 kg of $CO_2$ per gram into the atmosphere. On the other hand, the produced polypropylene releases approximately 1.9 kg of $CO_2$ per gram into the atmosphere [1]. These plastics are petrochemical-based materials extracted from petroleum oil. While the extraction and production of petroleum oil can produce 10.3 g of emissions for every megajoule of crude oil collected, countries with the most carbon-intensive practices produce emissions at nearly twice the rate [2]. Hence, conventional toothbrushes produced using plastics and nylon polymers have harmful effects on the environment.

In the height of climate change, concepts of energy saving, carbon reduction, environmental protection, and green materials have been developed. Bamboo is a giant grass and a non-timber forest resource used to create multiple products. Taiwan's bamboo resources possess advantages, including easy planting, rapid growth, rich yield, and ecological protection. Bamboo has the lowest energy requirement for production [3]. Thus, it can reduce the use of high energy-consumption building materials like concrete, bricks, cement, and steel and serve as a substitute solid wood material to reduce the pressure on forest resources. A bamboo-structure residential building prototype suggests that the bamboo-structure building requires less energy and emits less carbon [4]. Further, bamboo's carbon storage and sequestration rates are 30~121 Mg per ha and 6~13 Mg per ha per year, respectively [5]. In addition, bamboo (Phyllostachys pubescens) retains carbon longer than Chinese fir [6]. Zhu-Shan, one of the important towns of the bamboo industry in Taiwan, is famous for its excellent bamboo-processing industry. Its export is booming, and the industry is prosperous. At its peak, there have been more than 500 bamboo workshops in the town. However, the bamboo-processing industry in the town has declined in recent years due to the shift of industrial technology, meager profit, lack of youth input, high labor cost, declining competitiveness, and interruption of industrial and technical inheritance.

Currently, the town of Zhu-Shan is dominated by small-scale processing factories, and most of the processing technology is still in the preliminary stage. Yuan Peing Co., Ltd. (Yuan-Tai Bamboo Arts Club), located in the town, aims to develop and promote bamboo products as daily necessities. A toothbrush is an oral hygiene instrument used daily to clean teeth. In particular, bamboo toothbrush, a low-carbon lifestyle product, is the company's leading product. The plant-based bamboo toothbrush comprises a handle and bristles. The handle is made of bamboo, and the bristles are made of horse mane. This bamboo toothbrush is eco-friendly and biodegradable, which naturally decomposes when thrown away. The manufacturing process of bamboo toothbrush handles is a series of bamboo-shaped procedures. These procedures require multiple labor costs and woodworking machines, which entails many shortages in producing bamboo toothbrush handles. Nevertheless, the product's competitiveness can be increased by improving the process technology and product quality of the bamboo toothbrush handle.

With the rapid development of modern industrial equipment technology, there is an increasing demand for automation and intelligence. Not only can it increase production capacity and product yields and reduce labor costs, but it can also achieve intelligent control, data statistics, quality analysis, and remote monitoring. In addition, the equipment introduces a human–machine interactive interface, which assists operators in data collection and integration to reduce operators' mistakes in manual operation. The human–machine interactive interface allows operators to communicate with the machine, through which

the process data and information are recorded to allow operators to immediately diagnose the current status of the product or equipment.

Furthermore, this research aims to develop a bamboo toothbrush handle machine to optimize process conditions by a human–machine interactive interface. Although bamboo is a sustainable green material, the shaping process is difficult due to an extremely anisotropic property in bamboo structures [7,8]. Notably, the bamboo structure is more heterogeneous than that of wood [9,10]. The bamboo tissue can be regarded as a composite of vascular bundles embedded in a matrix of parenchyma cells. The vascular bundles are constructed from hollow vessels surrounded by fibrous sclerenchyma cells. The volume fraction of vascular bundles increases toward the outer part of the culm, which results in a pronounced radial density gradient with various mechanical properties in the bamboo culm [11–13]. The mechanical properties of Moso bamboo show significant variation with the radial position. Axial properties increase linearly with density, while the transverse compressive strength shows little variation. The radial density gradient of Moso bamboo varies from about 580 to 900 $kgm^{-3}$ for specimens taken from the inner to outer culm, which has a profound effect on the Moso bamboo's mechanical properties [7–17]. The axial tensile Young's modulus varies from about 5 to 25 GPa, and the axial tensile strength varies from about 100 to 800 MPa for specimens taken from the inner to outer culm [11–13]. The sclerenchyma fibers possess axial elasticity, while the axial strength has a significant contribution from the parenchyma matrix. Moreover, the mechanical properties of bamboo sections decrease with the increase in the wall thickness of the culm, which is associated with the reduced volumetric ratio of cellulose fibers to lignin as the culm diameter increases [15]. Bamboo is a sound structural and engineering material used in housing due to its strength, flexibility, and versatility [18]. Structural bamboo, as a substitute for structural timbers, can reduce the pressures on the shrinking natural forests and facilitate the conservation of the global environment [8]. Compared with North American softwoods, Moso bamboo is stronger, approximately as stiff, but significantly denser. However, the density and strength will likely present difficulties in the processing of Moso bamboo [14]. Therefore, it is important to optimize process conditions during the processing and shaping of bamboo to ensure that bamboo toothbrush handles with high quality can be stably mass-produced. The feed and spindle drive motor currents are often used to detect tool failure and estimate the cutting force in CNC machines [16,17]. In addition, displacement sensors are used to monitor the workpiece quality using a spindle-integrated high-resolution force measurement. The experimentally determined transfer function between the force and the displacement sensor can be employed to calculate the cutting forces during machining [19]. Furthermore, the application of micro strain gauges in small notches on the slide improves the sensitivity by integrating sensors into the slide for sensing the axis-slide in a manufacturing environment [20]. The measured cutting forces are further used to reconstruct the surface topographies of peripheral milled surfaces by the proposed tool model. The tool can obtain the accurate shape and roughness of machined surfaces [21]. Accurate cutting-force measurements are considered key information in machining that paves the way for understanding the cutting processes, optimizing the cutting operations, and evaluating the presence of instabilities related to the effectiveness of cutting processes [22]. Hence, the bamboo structure is an anisotropic property that can cause a rough or scorched surface and a tearing crack in bamboo toothbrush handles under improper process conditions. It is necessary to precisely control process conditions for retaining high-quality bamboo toothbrush handles in mass production.

The bamboo toothbrush handle milling machine is usually designed by a profiler [23], which uses various molds to change the shapes and sizes of bamboo toothbrush handles. This machine cannot probe the accurate cutting force for optimizing the cutting operations, paths, and parameters. A proper process condition can be probed using a human–machine interactive interface integrated with a CNC machine. The proposed machine is designed as a double-group to stably produce bamboo toothbrush handles in large quantities, which includes two storage racks of raw materials, two feeding devices, two exchange clamp-

ing devices, and a dual-spindle milling system necessary to form the shaping process of bamboo toothbrush handles. A computer numerical control (CNC) SYNTEC controller is adopted to propel the whole system and further integrate a human–machine interactive interface programmed by the LabVIEW software via a Modbus RTU communication protocol. The optimal milling paths, manufacturing methods, and feeding rates are verified using the surveillance system of the human–machine interactive interface to detect the instant currents of both spindles via the trial-and-error method and mass production. Hence, it can mass-produce high-quality bamboo toothbrush handles through the automation and intelligence system, reducing labor and production costs, decreasing the selling price, and increasing consumers' willingness to use and purchase bamboo toothbrushes.

## 2. Materials and Methods

The general process of bamboo toothbrushes includes six steps, as shown in Figure 1. First, bamboo is split into strips. Second, bamboo strips are flattened with a dimension of 184 (length) × 16 (width) × 5.5 (height) (mm) using a planer. Third, a profiler is used to form the toothbrush shapes by clamping and milling twice. Fourth, a polisher is used to grind rough edges changed into smooth corners. Fifth, water-based paint is used to spray on the bamboo toothbrush handles to prevent mildew. Finally, a driller is used to bore small holes at the head of bamboo toothbrush handles, and then a tufting machine is employed to implant horse manes at the holes. In these processes, both the third and the fourth steps are critical steps to deciding the profile of bamboo toothbrushes. Hence, a bamboo toothbrush handle machine is developed to form the profile of bamboo toothbrushes and optimize process conditions with a human–machine interactive interface. It can work on an extremely anisotropic property in bamboo structures to diminish a rough or scorched surface and a tearing crack, while Moso bamboo is used to fabricate bamboo toothbrush handles. The rough and scorched surfaces and the tearing cracks in bamboo toothbrush handles are illustrated in Figure 2.

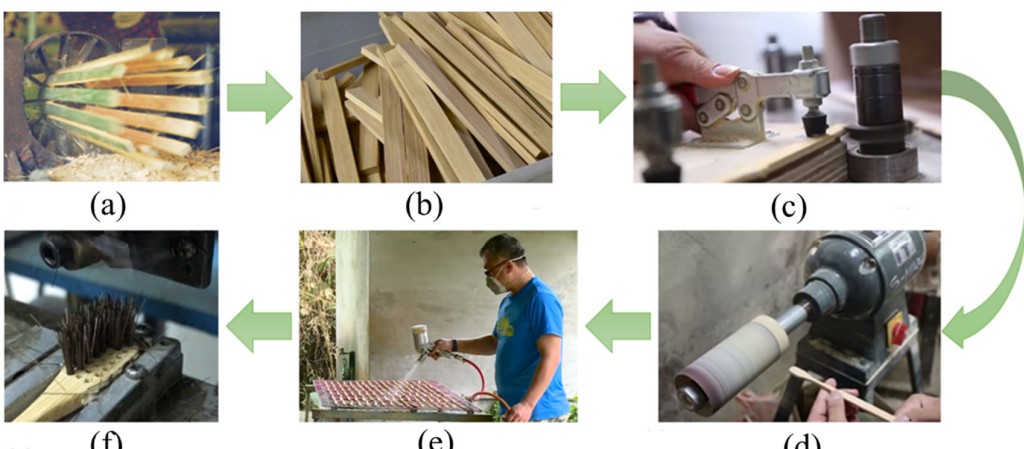

**Figure 1.** The general process of bamboo toothbrushes: (**a**) bamboo is split into strips; (**b**) bamboo strips are flattened using a planer; (**c**) a profiler is used to form the toothbrush shapes by clamping and milling twice; (**d**) a polisher is used to grind rough edges changed into smooth corners; (**e**) water-based paint is used to spray on the bamboo toothbrush handles to prevent mildew; (**f**) a driller is used to bore small holes at the head of bamboo toothbrush handles, and a tufting machine is employed to insert horse manes into the holes.

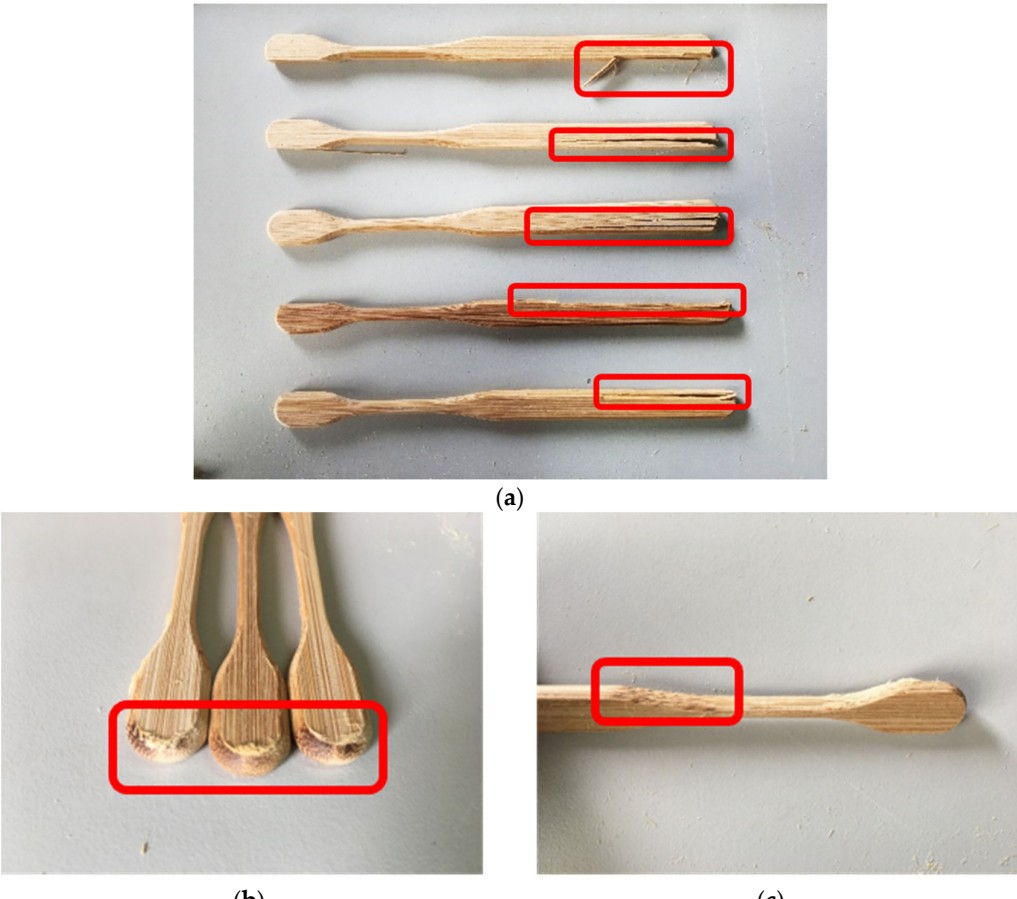

**Figure 2.** The damages on bamboo toothbrush handles under inadequate process conditions: (**a**) tearing cracks, (**b**) scorched surfaces, and (**c**) rough surfaces.

The bamboo toothbrush handle machine comprises two storage racks of flattened bamboo strips, two feeding devices, two exchange clamping devices, and a dual-spindle milling system to stably fabricate the handles with mass production via a double-group design, as shown in Figure 3. The process flow is shown in Figure 4. A proximity sensor is used to detect the stock of flattened bamboo strips in the storage rack. Then, a pneumatic cylinder is used to push the flattened bamboo strip forward. The feeding device is employed to clamp the flattened bamboo strip and put the strip on the work table. Simultaneously, one clutch of the exchange clamping device is operated to hold down the flattened bamboo strip on the work table. Then, the feeding device returns to the initial point. The dual-spindle milling system is operated to fabricate one side of the bamboo toothbrush handle. Another clutch of the exchange clamping device holds down the flattened bamboo strip on the work table, in which the dual-spindle milling system is also used to fabricate another side of the bamboo toothbrush handle. Finally, the fabricated bamboo toothbrush handle is released from the exchange clamping device and pushed into an unloading box by a biaxial pneumatic cylinder to complete the manufacture of the bamboo toothbrush handles.

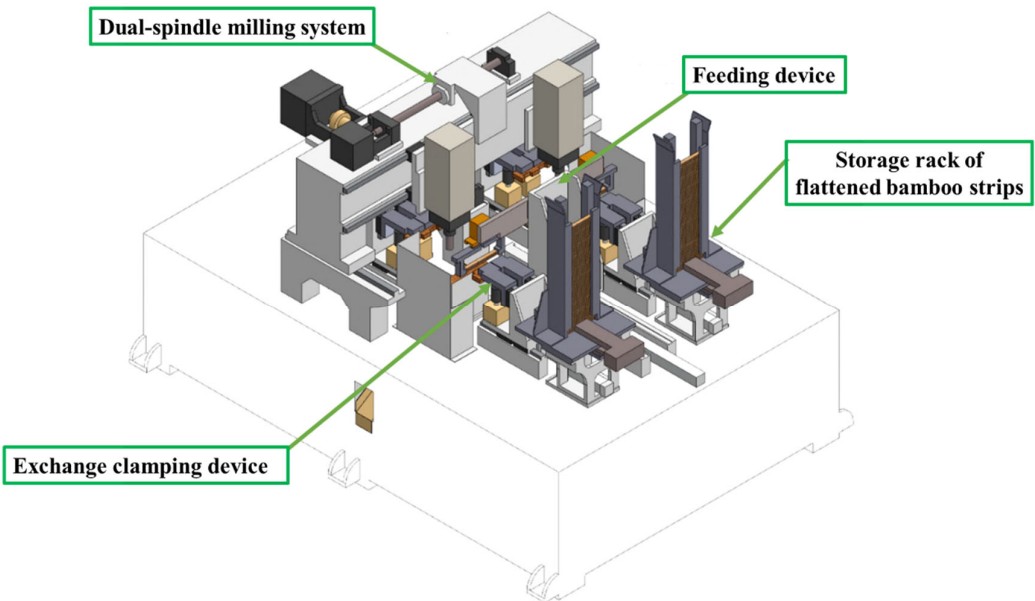

**Figure 3.** Schematic diagram of the bamboo toothbrush handle machine.

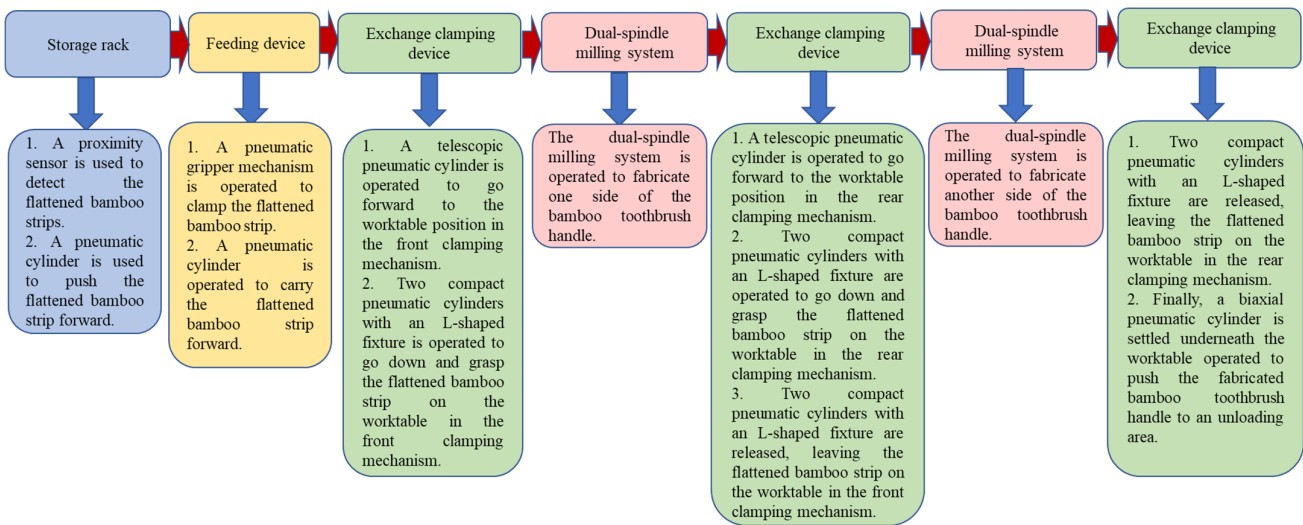

**Figure 4.** Process flow of the bamboo toothbrush handle.

The storage rack is filled with flattened bamboo strips by manual feeding. Then, the flattened bamboo strips fall vertically down to the bottom of the storage rack. The proximity detector (Model: PA18CSD04NASA) is used to sense the stock of the flattened bamboo strips in the storage rack. The telescopic pneumatic cylinder (Model: SC-32x200-S) connected with an aluminum plate by the spherical universal joint (Model: F-M12x125U) is used to push the flattened bamboo strip away from the storage rack to the reclaiming position, as shown in Figure 5. Subsequently, the feeding device is operated to clamp the flattened bamboo strip and put the strip on the work table. The feeding device comprises a pneumatic gripper mechanism, a pneumatic cylinder acting in a horizontal direction, and a pneumatic cylinder acting in a vertical direction, as shown in Figure 6. The actuation flow of the feeding device includes eight steps, as follows:

(1) The pneumatic cylinder acting in a vertical direction is operated to go down to the reclaiming position;

(2) The pneumatic gripper mechanism is operated to clamp the flattened bamboo strip;

(3) The pneumatic cylinder acting in a vertical direction is operated to go up to the initial position;

(4) The pneumatic cylinder acting in a horizontal direction is operated to carry the flattened bamboo strip forward;

(5) The pneumatic cylinder acting in a vertical direction is operated to go down to the work table position;

(6) The pneumatic gripper mechanism releases and leaves the flattened bamboo strip on the work table;

(7) The pneumatic cylinder acting in a vertical direction is operated to go up to the initial position;

(8) The pneumatic cylinder acting in a horizontal direction is operated to return to the initial position.

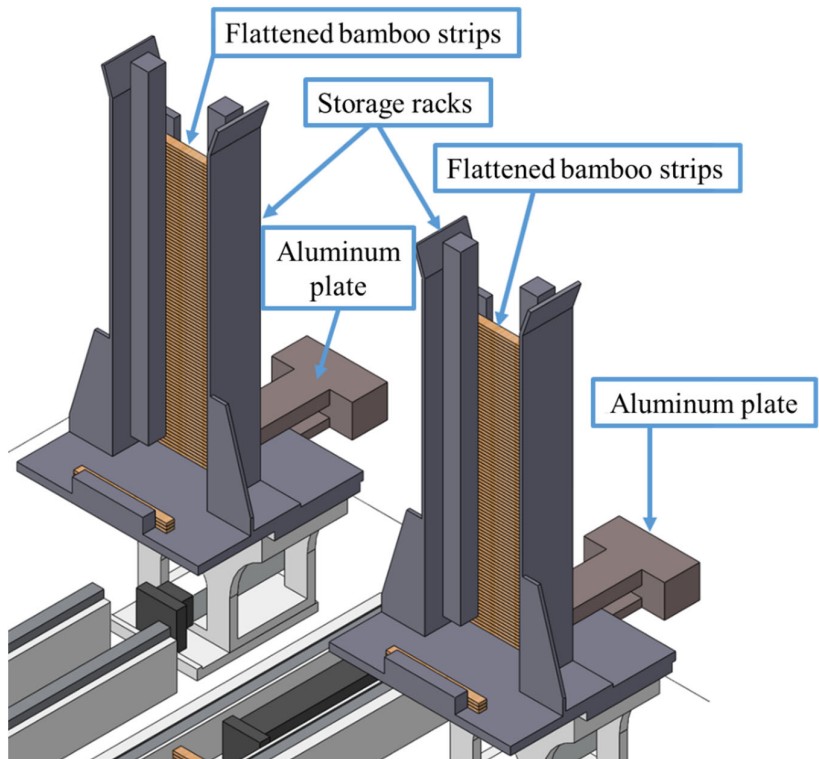

**Figure 5.** Schematic diagram of the storage racks of the flattened bamboo strips.

The exchange clamping device is used to successively hold down the flattened bamboo strip on the work table to further fabricate both profiles of bamboo toothbrush handles using the dual-spindle milling system, as shown in Figure 7. The exchange clamping device comprises a front and rear clamping mechanism and a work table. The front clamping mechanism is almost the same as the rear clamping mechanism. The fixture shape in both mechanisms is a complementary concave-convex design that matches the operation of exchange clamping. The front or rear clamping mechanism comprises two floating connectors (Model: F-M18x150F), two compact pneumatic cylinders (Model: ACQ63x30S) acting in a vertical direction, four linear guides and sliders (Model: MSA25S), a floating connector (Model: F-M12x125F), a telescopic pneumatic cylinder (Model: SC40x150-S-S2) acting in a horizontal direction, and an L-shaped fixture. The L-shaped fixture is connected with two linear guides and sliders (Model: MSA25S), which are actuated by two compact pneumatic cylinders (Model: ACQ63x30S) with two floating connectors (Model: F-M18x150F) that grip the flattened bamboo strip on the work table. These devices are connected with two linear guides and sliders (Model: MSA25S), which are actuated by the telescopic pneumatic cylinder (Model: SC40x150-S-S2) with the floating connector (Model: F-M12x125F) that help locate the grasp position on the flattened bamboo strip. The actuation flow of the exchange clamping device includes eight steps, as follows:

(1) In the front clamping mechanism, the telescopic pneumatic cylinder acting in a horizontal direction is operated to go forward to the work table position;

(2) The two compact pneumatic cylinders with the L-shaped fixture are operated to go down to grasp the flattened bamboo strip on the work table;

(3) The dual-spindle milling system is actuated to fabricate one side of the bamboo toothbrush handle;

(4) In the rear clamping mechanism, the telescopic pneumatic cylinder acting in a horizontal direction is operated to go forward to the work table position;

(5) The two compact pneumatic cylinders with the L-shaped fixture are operated to go down and grasp the flattened bamboo strip on the work table. At this time, both L-shaped fixtures in the front and the rear clamping mechanism are simultaneously operated to grasp the flattened bamboo strip on the work table. The L-shaped fixtures in both mechanisms are a complementary concave-convex design favoring the operation of exchange clamping;

(6) In the front clamping mechanism, the two compact pneumatic cylinders with the L-shaped fixture release and leave the flattened bamboo strip on the work table. The telescopic pneumatic cylinder acting in a horizontal direction is operated to return to the initial position;

(7) The dual-spindle milling system is actuated to fabricate another side of the bamboo toothbrush handle;

(8) In the rear clamping mechanism, the two compact pneumatic cylinders with the L-shaped fixture release and leave the flattened bamboo strip on the work table. The telescopic pneumatic cylinder acting in a horizontal direction is operated to return to the initial position;

(9) Finally, a biaxial pneumatic cylinder settled underneath the work table is operated to push the fabricated bamboo toothbrush handle to an unloading area.

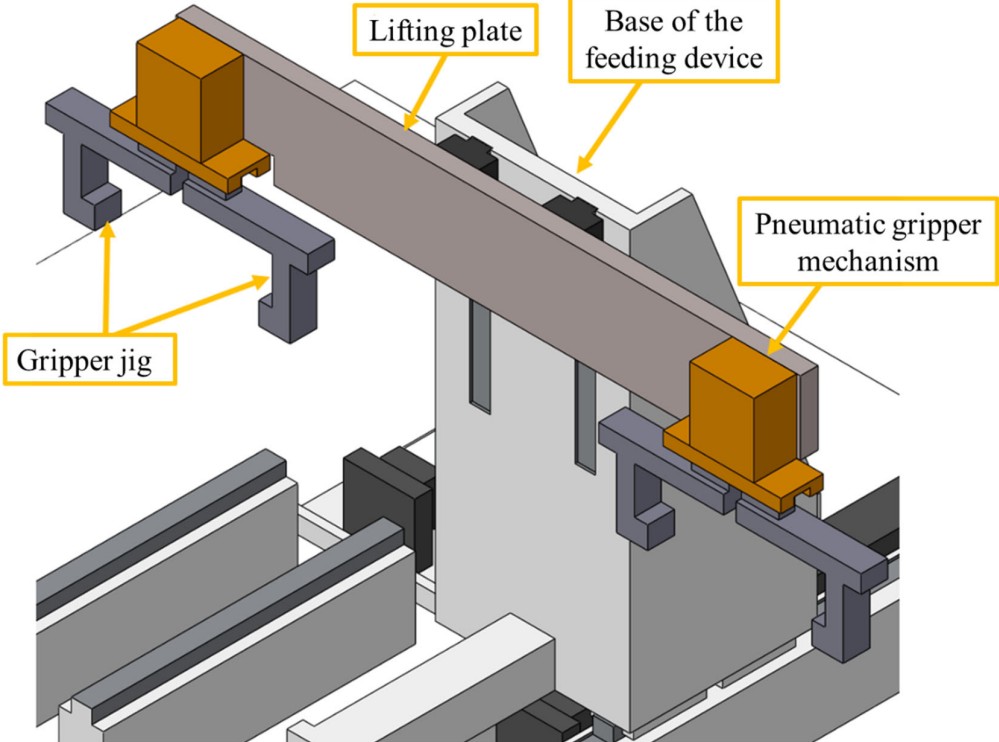

**Figure 6.** Schematic diagram of the feeding device used to carry the flattened bamboo strips from the reclaiming position to the work table position.

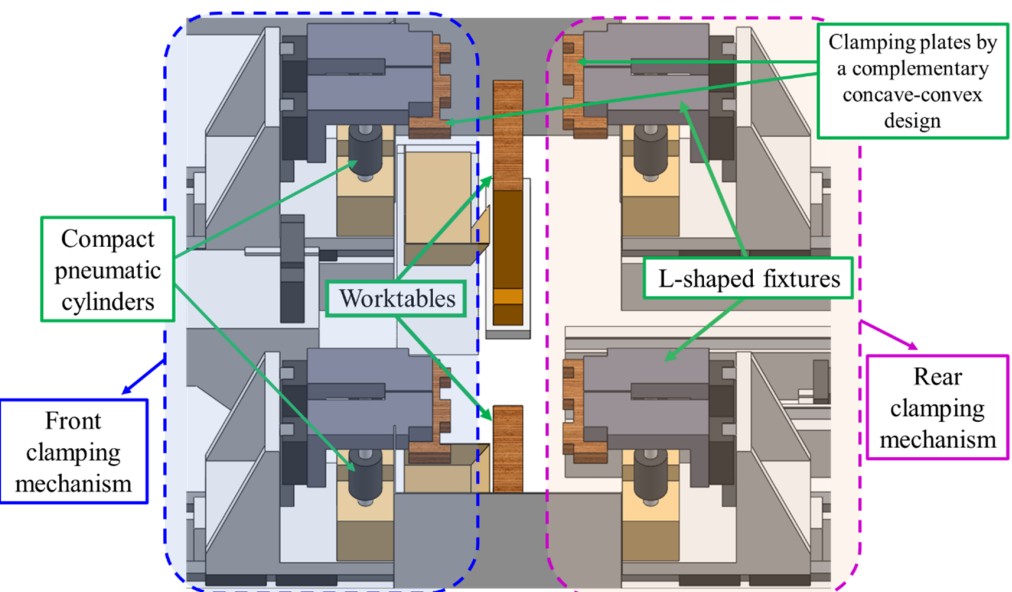

**Figure 7.** Schematic diagram of the exchange clamping device used to successively hold down the flattened bamboo strip on the work table and fabricate both profiles of bamboo toothbrush handles using the dual-spindle milling system.

The dual-spindle milling system comprises two spindles (Model: AW14-K8-S18-ER32), an AC servo motor (Model: SME-M15020SCB) acting in an X-axis direction (left-right direction) with a retainer (Model: MBCS25-G), a transport ball screw (Model: FSDC03210T5-D) and a bearing seat (Model: BF30.6206), an AC servo motor (Model: SME-M20020SCB) acting in a Y-axis direction (front-rear direction) with a retainer (Model: MBB30H), a transport ball screw (Model: FDICR4010T4), a bearing seat (Model: BF30.6206), a ball screw support unit (Model: WBK30DF-TPI) and a coupling (Model: SAP-94C-25-35), two linear guides with four sliders (Model: MSA30E), a C-shaped steel frame, and an L-shaped steel frame, as shown in Figure 8. The L-shaped steel frame is used to connect two spindles to achieve a consistent motion along the X-axis direction. The C-shaped steel frame is adopted to construct the main structure and brace the X-axis transmission mechanism, which is driven along the Y-axis direction by the Y-axis transmission mechanism.

An electrical control system is adopted to drive the dual-spindle milling system, the feeding device, the exchange clamping device, and the storage rack, which comprises a computer numerical controller CNC (SYNTEC, Model: 11MA), a servo driver (Model: SDE-150A2) for the X-axis servo motor, a servo driver (Model: SDE-200A2) for the Y-axis servo motor, two inverters (Model: AW14-K8-S18-ER32) for controlling the rotational speed of both spindles and relay module for driving the pneumatic components (shown in the Appendix A). The specification of the computer numerical controller is shown in Table 1. The specification of the servo drivers for propelling the servo motors is shown in Table 2, and the specification of the AC servo motors is shown in Table 3. The specification of the inverter for propelling the spindle is shown in Table 4, and the specification of the spindle is shown in Table 5.

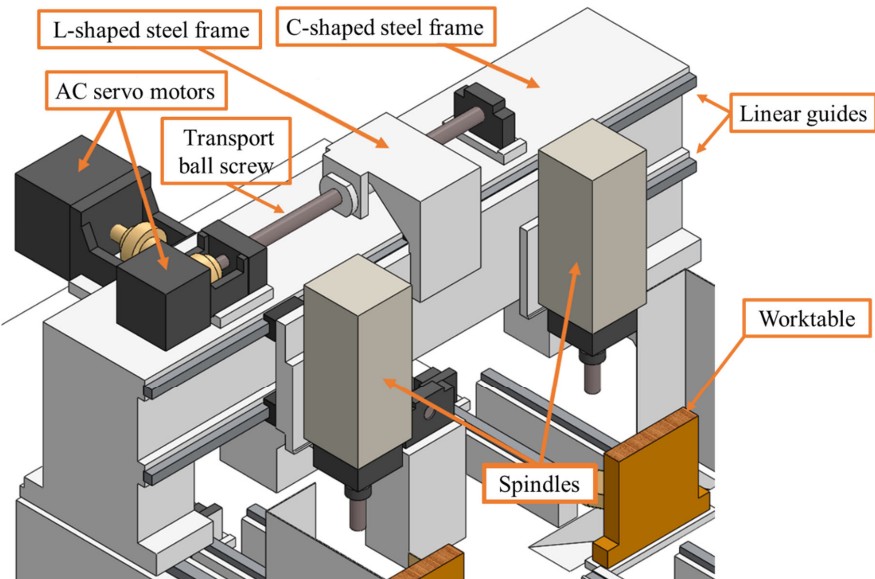

**Figure 8.** Schematic diagram of the dual-spindle milling system used to fabricate the profile of bamboo toothbrush handles.

**Table 1.** The specification of the computer numerical controller.

| CNC controller | SYNTEC (Model: 11MA) |
|---|---|
| Number of axis | 4 |
| Spindle group | 2 |
| Panel | 10.4 inch |
| Storage space | 4 GB |
| Number of I/O | 32 |
| Communication mode | RS-232, RS-485, Ethernet |

**Table 2.** The specification of the servo drivers.

| Servo driver | SDE-150A2 | SDE-200A2 |
|---|---|---|
| Phase | Single-phase or three-phase | |
| Encoder | 22 Bit | |
| Current | 9.4 A | 12.1 A |
| Frequency | 250 Hz | |
| Power | 30 W | |

**Table 3.** The specification of the AC servo motors.

| Servo motor | SME-M15020SCB | SME-M20020SCB |
|---|---|---|
| Rated power | 1500 W | 2000 W |
| Rated torque | 7.16 N-m | 9.55 N-m |
| Maximum torque | 21.6 N-m | 28.5 N-m |
| Rated current | 8.5 A | 11 A |
| Maximum current | 25.2 A | 34.7 A |
| Rated Speed | 2000 RPM | |
| Maximum Speed | 3500 RPM | |
| Encoder | 22 Bit | |

Table 4. The specification of the inverter.

| Inverter | MS300-11KW-15HP-220V |
|---|---|
| Phase | Three-phase, 230 V |
| Motor power | 11 KW, 15 HP |
| Rated current | 49 A, 51 A |
| Maximum output frequency | 599 Hz |
| Communication mode | MODBUS TCP/IP, Ethernet |

Table 5. The specification of the spindle.

| Spindle | AW14-K8-S18-ER32 |
|---|---|
| Maximum frequency | 600 Hz |
| Rated frequency | 400 Hz |
| Rated voltage | 220 V |
| Mean frequency | 200 Hz |
| Mean Voltage | 110 V |
| Minimum output frequency | 2 Hz |
| Minimum output voltage | 1.1 V |
| Upper limit frequency | 95% |
| Lower limit frequency | 5% |

The pneumatic system in the bamboo toothbrush handle machine is equipped with ten sets of pneumatic cylinders. The 5/2-way-solenoid valve (Model: MVSC-300-4E2P-AC220) is used to control the blow function in the spindle and remove the chips of bamboo during processing. The 5/2-way-solenoid valve (Model: MVSC-300-4E2C-AC220) is adopted to control the pneumatic cylinder acting in a horizontal direction to carry the flattened bamboo strips forward and backward in the feeding device and the telescopic pneumatic cylinders acting in a horizontal direction in the front and the rear clamping mechanism. The 5/2-way-solenoid valve (Model: MVSC-300-4E1-AC220) is used to control the pneumatic gripper mechanism, the pneumatic cylinder acting in a vertical direction in the feeding device, and the compact pneumatic cylinders acting in a vertical direction in the front and the rear clamping mechanism.

The communication settings are adopted to establish a communication connection between the computer and the controller of SYNTEC 11MA. The communication interface of RS-485 is used to further set the SYNTEC controller as a master end and the computer as a slave end by a Modbus RTU communication protocol. The data transmission uses hexadecimal which is a base-16 number system with 16 possible digits used to represent numbers. The communication component addresses of CNC SYNTEC 11MA use the registers to read or write communication data. The communication PLC program must first set the number and the address of the registers. The communication component addresses of R801~R822 in the controller of SYNTEC 11MA are used to correspond with those of 0000H~0021H in the human–machine interactive interface computer. The human–machine interactive interface uses the Modbus RTU communication protocol to connect the computer and the controller through the PLC software (Ladder Programming, LAD) and LabVIEW (Visual programming language, VPL). The flow chart of the communication program is shown in Figure 9. The communication module can supply three modes—ASCII, RTU, and TCP/IP—in the communication protocol of LabVIEW Modbus. Three functional elements, namely Create Serial Slave VI, Read Holding Registers VI, and Shutdown VI, are employed to compile the communication program in LabVIEW. The functional element of Create Serial Slave VI is mainly used to set the communication parameter data with

the communication equipment. In this study, some parameters are set to communication mode as RTU, data transfer rate as 9600(bps), slave terminal number as 1, data content length as 8 bits, stop bit as 1 bit, parity check as none, and selecting a communication port (COM port) to implement data transmissions between the computer and the controller. The functional element of Read Holding Registers VI is adapted to read from the start address of the equipment and write data to the registers. Then, the functional element of Shutdown VI is used to forcibly terminate the current communication data stream and close the whole communication connection.

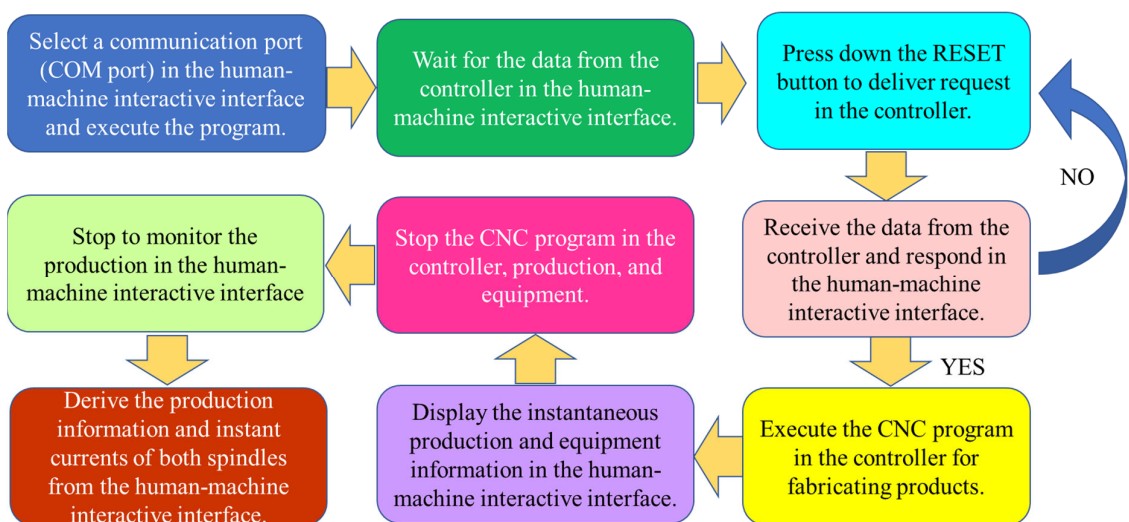

**Figure 9.** The flow chart of the communication program between the computer and the controller.

There are six function screens, namely the home page, menu, process, mobile version, monitoring, and recording in the design of the human–machine interactive interface computer. The product photo of bamboo toothbrushes is used to construct the background of the home page with the trademark of Yuan Peing Co., Ltd. and the school badge of National Formosa University. In this screen, a communication port (COM port) must be selected to further click the button for executing the human–machine interactive interface computer, as shown in Figure 10. Then, it enters the menu screen with three options: process, mobile version, and monitoring buttons, as shown in Figure 11. It shows the process detail information by an animated interactive interface while clicking the process button. Additionally, it shows brief information about the process for tracking productivity and the utilization rate after clicking the mobile version button. Subsequently, it displays the instant current of both spindles by waveform diagrams after clicking the monitoring button. Moreover, it shows a record involving the date, time, overcurrent values, and rotational speeds of both spindles after clicking the recording button on the monitoring screen.

Twelve items are displayed on the process screen with animated graphs and instant values to present the processing status. The lateral view of the machine as the background is used to illustrate the equipment operation, as shown in Figure 12. These items include:

(1) The status of the storage racks presented by an animate graph;
(2) The number of flattened bamboo strips in the storage racks presented by a bar light;
(3) The status of the feeding device presented by an animate graph;
(4) The status of the dual-spindle operation presented by an animate graph;
(5) The status of equipment operation is presented by three lights: green, yellow, and red. The green light indicates machining. The yellow light indicates a temporary stop during processing. The red light indicates an operation error or insufficient raw material preparation;
(6) The status of the exchange clamping device presented by an animate graph;
(7) The amount of unloading on both sides;
(8) The amount of raw material in the storage racks;

(9)  The amount of intraday production capacity;
(10)  The number of finished products calculated from the first production to the production;
(11)  The rotational speed of both spindles;
(12)  The feeding rate for the x-axis and the y-axis motion.

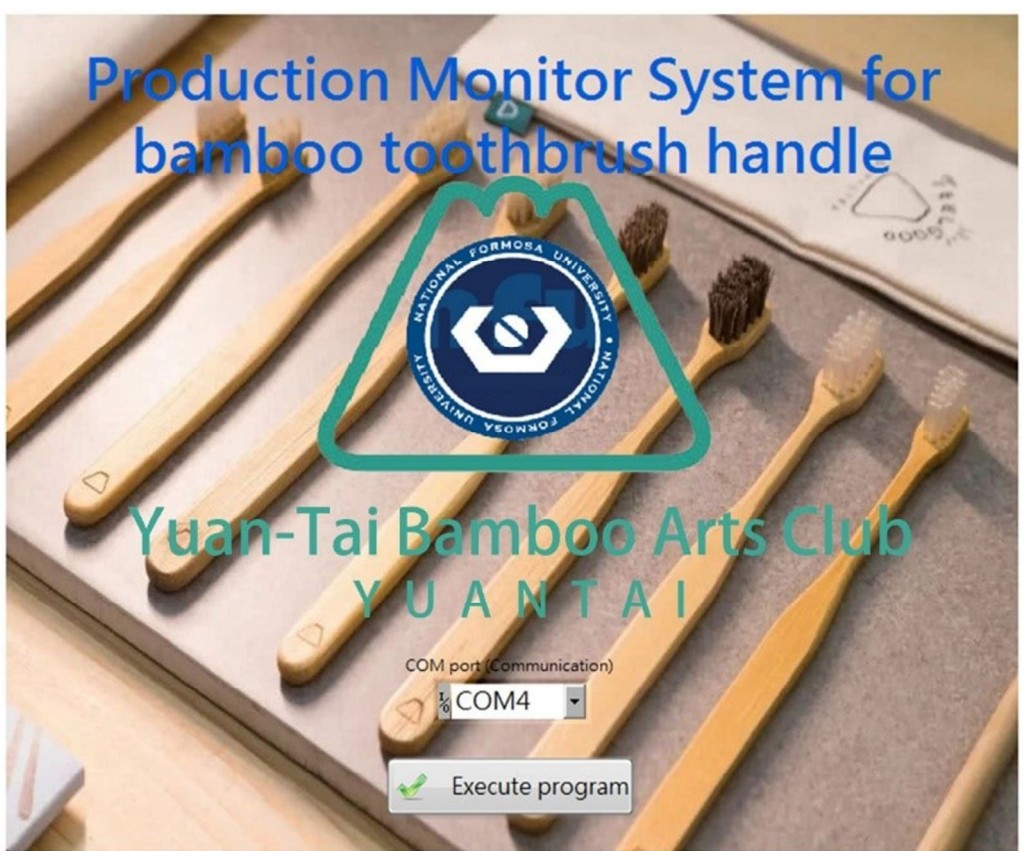

**Figure 10.** The human–machine interactive interface computer on the home page screen.

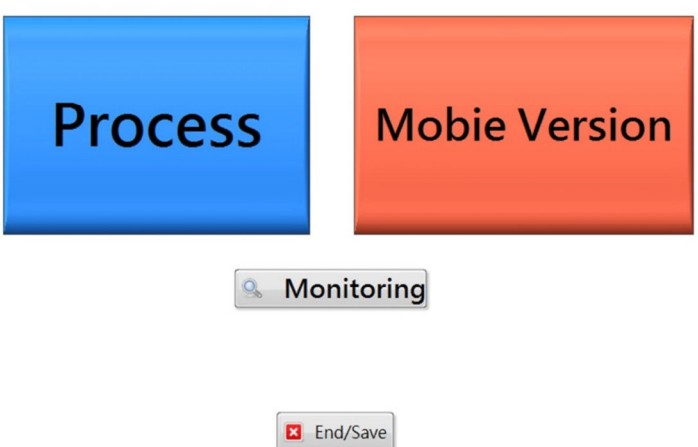

**Figure 11.** The human–machine interactive interface computer on the menu screen.

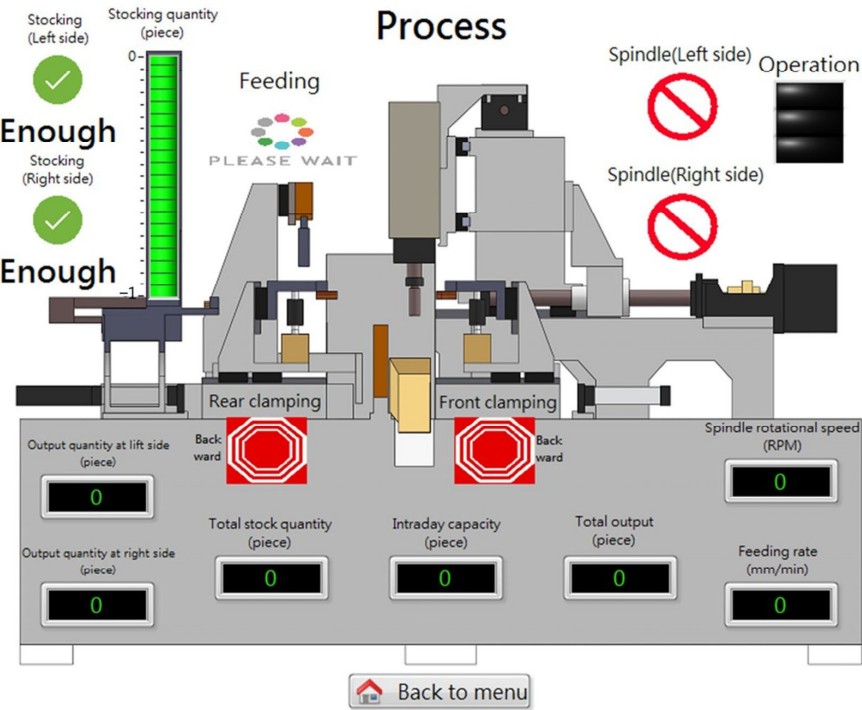

**Figure 12.** The human–machine interactive interface computer on the process screen.

The mobile version screen is designed for production management—to quickly track productivity and utilization rate. There are six items displayed by a bar light and instant values to present the processing status, as shown in Figure 13. These items include:

(1) The amount of cumulative production;
(2) The amount of raw material in the storage racks;
(3) Equipment operating time;
(4) Current time;
(5) The status of production progress presented by a bar light;
(6) The amount of current production.

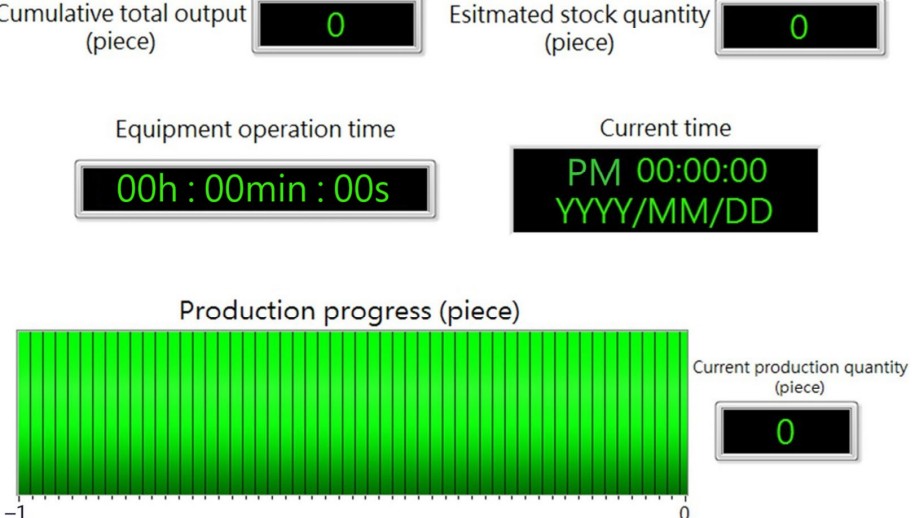

**Figure 13.** The human–machine interactive interface computer on the mobile version screen.

The monitoring screen is designed to monitor the instant current of both spindles by two waveform diagrams, as shown in Figure 14. The instant current is extracted from the inverter, which is used to control the rotational speed of the spindle and reflect the machining resistance. Meanwhile, a formed milling cutter is used to fabricate the bamboo toothbrush handle. It is designed as a semicircular blade to fabricate a semicircular corner along the edge of the bamboo toothbrush handle, as shown in Figure 15. The increased current can explain why the machining resistance increases to probably induce a rough or scorched surface to further cause a tearing crack on the bamboo toothbrush handle. Therefore, an appropriate process condition can be estimated by monitoring the instant current of the spindle, which needs to decide some parameters, including the feeding rate at the straight or curve machining, the rotational speed of the spindle, and the milling path (down or up milling). Hence, a threshold value of the instant current is further decided to infer the machining resistance, whether the rough or scorched surface and the tearing crack will occur or not. The overload current is wholly recorded on the recording screen. The record data include the date, time, overcurrent values, and rotational speeds of both spindles, as shown in Figure 16. This information can infer the yield rates, abrasion of the milling cutter or machine, and properties of the raw material. The information in the process and recording screen is further saved as two files of EXCEL while the human–machine interactive interface computer is out of operation.

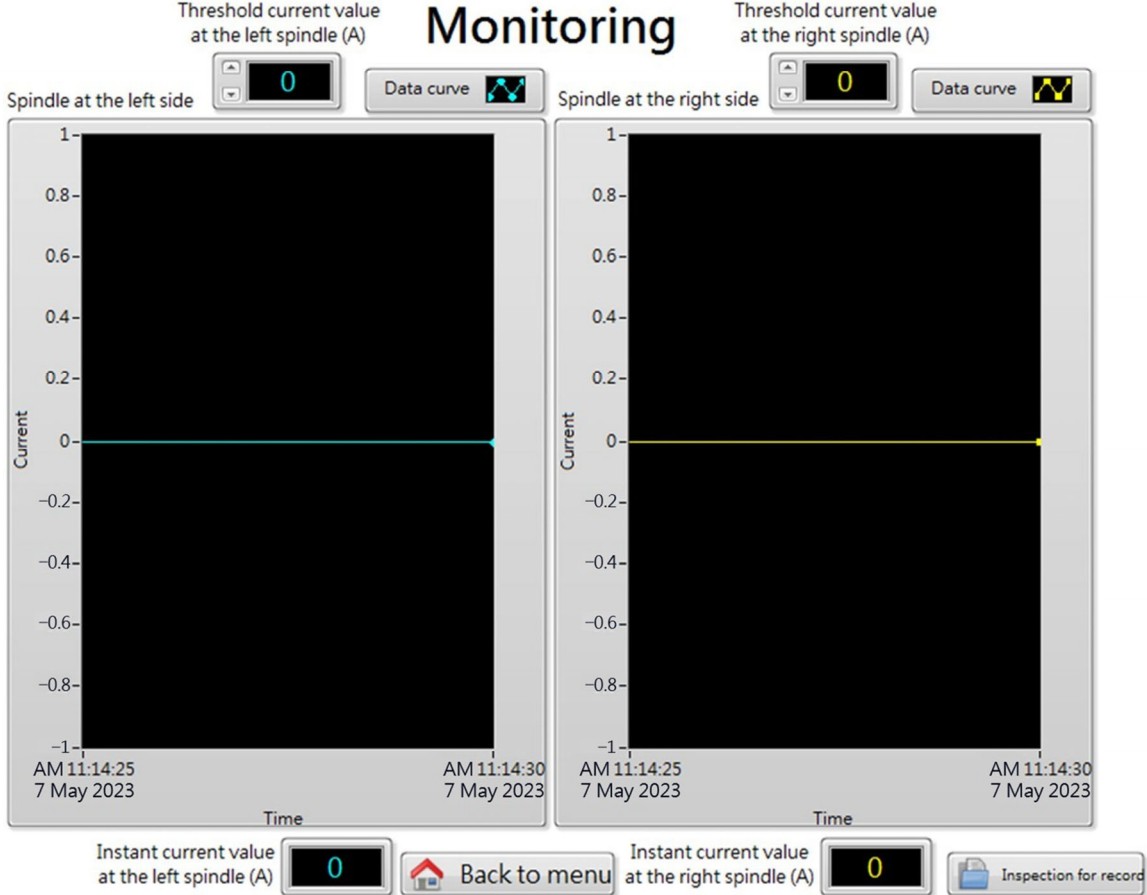

**Figure 14.** The human–machine interactive interface computer on the monitoring screen.

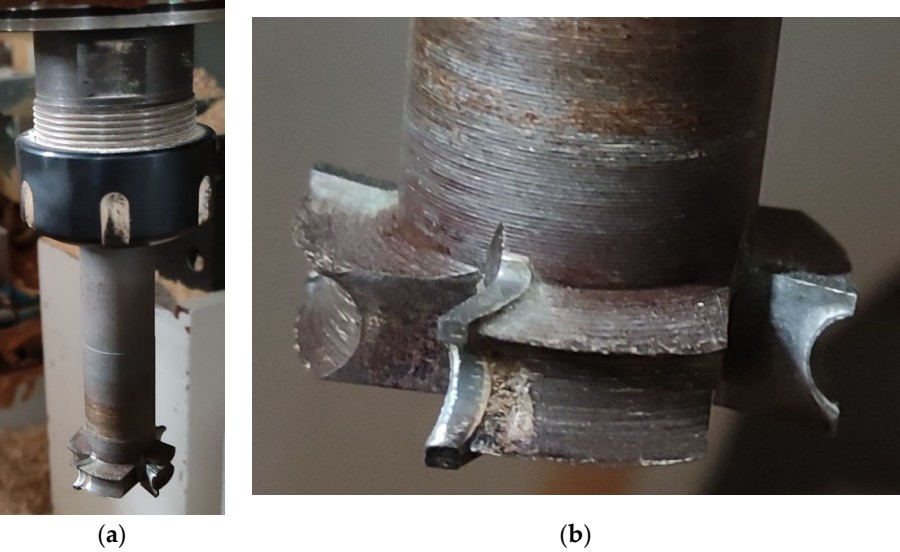

(**a**)                                               (**b**)

**Figure 15.** The formed milling cutter is used to fabricate a semicircular corner along the edge of the bamboo toothbrush handle: (**a**) an entire cutter and (**b**) an enlarged view of the front end of the cutter.

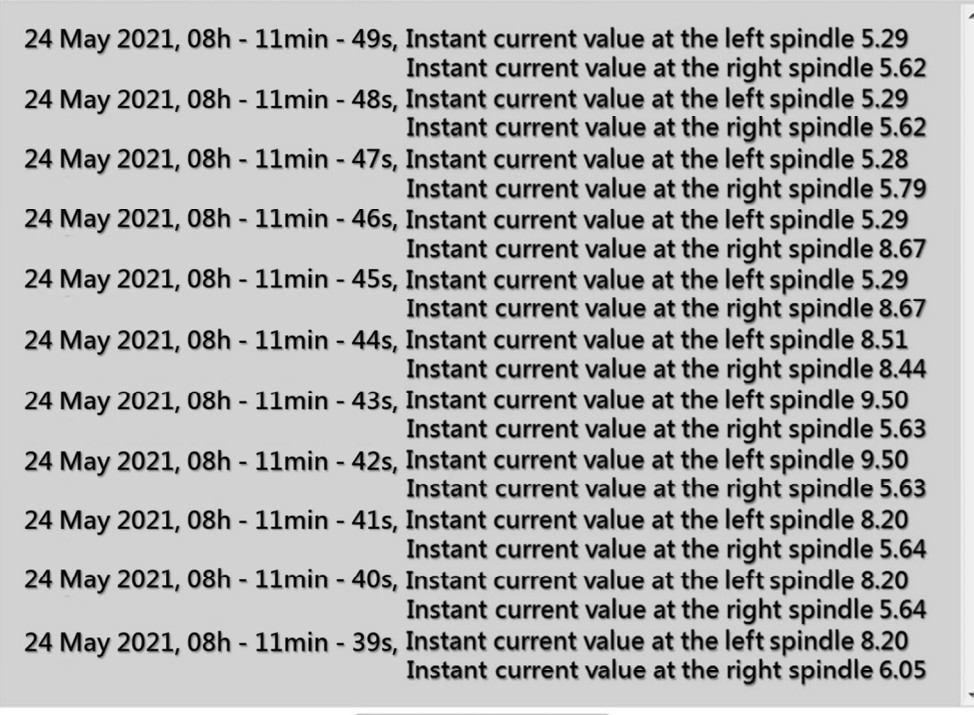

**Figure 16.** The human–machine interactive interface computer on the recording screen.

## 3. Results and Discussions

The climatic conditions of Taiwan have bred abundant bamboo species. In particular, previous generations have accumulated bamboo cultures and craftsmanship, further creating more bamboo craft techniques in Taiwan. These crafts representatively include [24]:

(1)   Bamboo tube cladding and drilling;
(2)   Butterfly-shaped rattangle;
(3)   Star weaving;
(4)   Square weaving;

(5)   Single-circle weaving;

(6)   Hexagonal weaving.

These techniques are used to fabricate products such as roof tiles, rafts, shoes, furniture, and agricultural tools, which are constructed on the bamboo properties of sturdiness and tenacity.

Bamboo is common in Chinese culture. Meanwhile, Taiwan's bamboo industry has suffered a rapid decline in recent decades due to people's preference for more modern-looking products or cheaper plastic goods and the impact of cheaper bamboo imported from China and Southeast Asia. This phenomenon is more obvious in the central part of Taiwan, namely Zhushan or the "Bamboo Mountain" town. In the past, people labor in this industry in every part of Taiwan. Over the years, bamboo has been replaced by plastic or steel to make all sorts of products. Bamboo has only been recently used to make items such as shampoo and insect repellent, socks, gloves, and even roasted peanuts. This reinvention has made a business opportunity for bamboo for small family businesses to complement the industry and reverse the decades of decline. Many of the manufacturers in Zhushan are breaking even but not making big profits. Nevertheless, they persevere to continue and promote the use of bamboo.

Ecological sustainability is a basic requirement for sustainable economic and social development. It is an obstacle to sustainable development for resulting pollutant emissions of material resources rather than consuming material resources [3]. Bamboo is a plant with fast growth. It only needs five years to reach maturity and is one of the most environmentally-friendly plants. People hope that bamboo can replace the use of other woods and plastics. Although plastic is very cheap and can be used for a longer time, bamboo is an environmentally friendly material. Moreover, bamboo grows without the necessity of fertilizers or pesticides. In addition, bamboo can also absorb a lot of carbon dioxide and reduce greenhouse gases. Taiwan cannot thrive without bamboo, which has been relied on for generations.

Bamboo is an abundant and high-throughput material. Hence, it has been integrated into the livelihood of people, which can be developed into an industry with more profits and business opportunities. Yuan Peing Co., Ltd. (Yuan-Tai Bamboo Arts Club) aims to develop and promote bamboo products. Although bamboo is a friendly material, it has extremely anisotropic properties. Therefore, it is important to introduce modern automation and an intelligent machine with optimal process conditions during the processing and shaping of bamboo, which can be used to stably mass-produce bamboo toothbrush handles with high quality. This machine can increase production capacity and product yield and reduce labor costs. Additionally, it can achieve intelligent control, data statistics, quality analysis, and remote monitoring.

The bamboo toothbrush handle milling machine is usually designed by a profiler [23], which uses various molds to change the shapes and sizes of bamboo toothbrush handles. This machine cannot probe the accurate cutting force for optimizing the cutting operations, paths, and parameters. The fabricated bamboo toothbrush handle machine is shown in Figure 17, which comprises a processing facility, a CNC controller, and a computer. The CNC controller is SYNTEC with Model: 11MA. The computer is used to construct the human–machine interactive interface programmed by LabVIEW software via a Modbus RTU communication protocol and connect the CNC controller. A double-group design is employed to stably produce bamboo toothbrush handles in large quantities in the processing facility. The flattened bamboo strip can store up to a maximum of 110 sticks in the storage rack, as shown in Figure 18. The proximity sensor is employed to sense the stock of the flattened bamboo strips and remind operators to execute stuffing. The telescopic pneumatic cylinder connected with the aluminum plate by the spherical universal joint is used to push one flattened bamboo strip forward to the reclaiming position at a time. Then, the feeding device is operated to clamp the flattened bamboo strip and put the strip on the work table. At the moment, the front clamping mechanism of the exchange clamping device is operated to hold down the flattened bamboo strip on the work table. The feeding

device returns to its initial position. The dual-spindle milling system is actuated to fabricate one side of the bamboo toothbrush handle. The rear clamping mechanism of the exchange clamping device proceeds to hold down the flattened bamboo strip on the work table. Subsequently, the dual-spindle milling system is also operated to fabricate another side of the bamboo toothbrush handle. These critical procedures of the fabricated mechanism are shown in Figure 19. The formed milling cutter with a semicircular blade is employed to fabricate a semicircular corner along the edge of the bamboo toothbrush handle. Then, the fabricated bamboo toothbrush handle is released from the exchange clamping device and pushed to an unloading box using a biaxial pneumatic cylinder to finish the manufacture of the bamboo toothbrush handle.

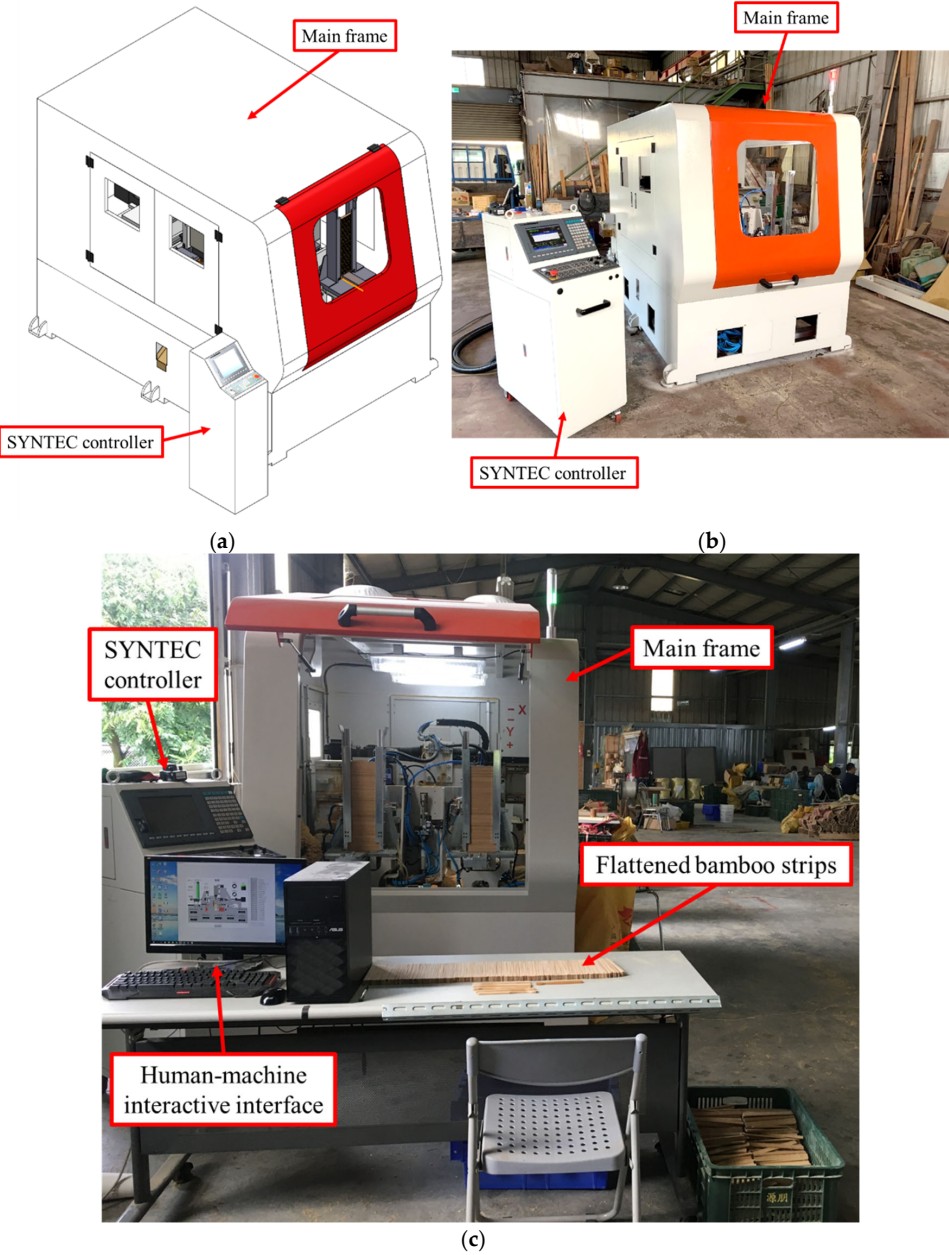

**Figure 17.** The fabricated bamboo toothbrush handle machine: (**a**) schematic diagram, (**b**) actual machine, and (**c**) entire structure.

The dual-spindle milling system is driven by the electrical control system. The inverter is used to drive the spindle within the rotational speed of 12000~18000 RPM. The linear motion of the spindles is achieved by two servo drivers for the X and the Y-axes servo

motors via the CNC controller (SYNTEC, Model: 11MA). The CNC controller can facilitate altering the processing path for modifying the profile of the bamboo toothbrush handles. Furthermore, the processing path is planned by the computer-aided manufacturing (CAM) software of SG CAM. The procedures to compile the processing path by SG CAM include five steps, as follows.

(1) The build cutter library is used to set the cutter parameters (feed rate and spindle speed) used in the shaping process;
(2) The set manufacturing method is used to set the path-planning and manufacturing method. The aspects of path-planning considerations involve a cut-in path, a moving path, and a cut-out path. The manufacturing methods involve down milling, up milling, and cutter compensation;
(3) Confirming the setting method is performed to check all processing methods and use various color line segments to distinguish the number and position of the setting methods;
(4) Performing machining simulation is employed to estimate the processing methods by simulating processing paths;
(5) Translating and producing G code are employed to generate the NC code according to the setting methods and processing paths. The generated NC code is imported into the CNC controller (SYNTEC, Model: 11MA) to execute the program and practice the machining of the bamboo toothbrush handles.

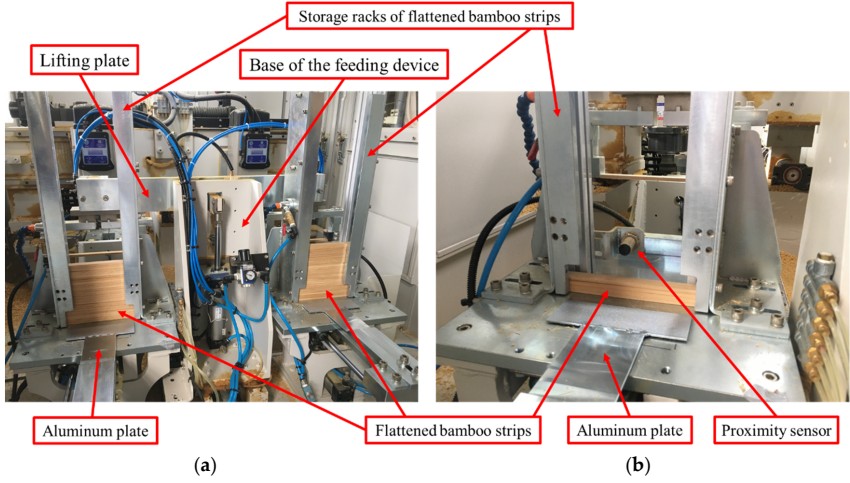

**Figure 18.** The fabricated storage racks of the flattened bamboo strips: (**a**) the entire structure and (**b**) a detailed configuration.

The production monitoring and optimizing process conditions are performed by the human–machine interactive interface computer, which uses the Modbus RTU communication protocol to connect the computer and the controller by the PLC software with the version number of GX-Works 2 (Ladder Programming, LAD) and LabVIEW (visual programming language, VPL). The process screen in production is shown in Figure 20, and then the monitoring screen in production is shown in Figure 21. The threshold current of both spindles is set as 8 A, which is estimated during mass production. While the instant current is over 8 A, the relevant information is recorded on the recording screen.

The monitoring and recording screens in the human–machine interactive interface computer are adopted to estimate the manufacturing methods and the path planning and further optimize process conditions. It cannot adopt a single manufacturing path along the profile of the bamboo toothbrush handles because the property of bamboo is an extremely anisotropic structure. This method designs six manufacturing paths with two manufacturing methods of down milling and up milling to accurately fabricate bamboo toothbrush handles. The detailed manufacturing paths and methods of the bamboo toothbrush handles are shown in Figure 22. The manufacturing method of down milling is applied to manufac-

turing paths 1, 3, and 6, and then the manufacturing method of up milling is applied to manufacturing paths 2, 4, and 5. The dotted lines present an empty cutting path with the G00 code in the NC coding. The solid lines present an actual cutting path with the codes G01, G02, or G03 in the NC coding. The manufacturing path is sequentially from 1 to 6. The starting point of the whole process is at the dotted line of cut-in path 1. The terminal point of the whole process is at the dotted line of the cut-out path 6. The whole manufacturing path is estimated and planned by mass production and the trial-and-error method. The manufacturing paths 1 and 2 are first performed while the front clamping mechanism of the exchange clamping device is operated to hold down the flattened bamboo strip on the work table. Then, manufacturing paths 3 to 6 are sequentially performed while the rear clamping mechanism of the exchange clamping device is operated to hold down the flattened bamboo strip on the work table. The instant current of the spindle is recorded along the manufacturing paths 1 to 6, which varies from 5.15 A to 5.35 A for the spindle on the left-hand side. Moreover, the instant current of the spindle on the right-hand side varies from 5.48 A to 5.68 A. The instant currents in a situation of empty cutting are 5.12 A and 5.45 A for both spindles on the left-hand and the right-hand side, respectively. The empty cutting is a standby itinerary for changing the front clamping mechanism to the rear clamping mechanism. The instant currents of both spindles have a slight difference with an acceptable range. This can attribute to some factors related to the installation and structure of the spindle. It is obvious that the instant current of the spindle is lower while the manufacturing path is a straight line. Furthermore, the instant current of the spindle is higher while the manufacturing path is a curve. The lowest instant currents of both spindles occur at the straight line of the manufacturing path 2. The highest instant current of the spindle on the left-hand side occurs at manufacturing paths 5 and 6. Further, the highest instant current of the spindle on the right-hand side occurs at the manufacturing paths 3, 5, and 6. The detailed processing conditions related to the manufacturing paths are shown in Table 6. After a long period of production and verification, this machine can produce more than 1600 bamboo toothbrush handles per day. The production information can be output as an Excel file while the human–machine interactive interface is out of operation.

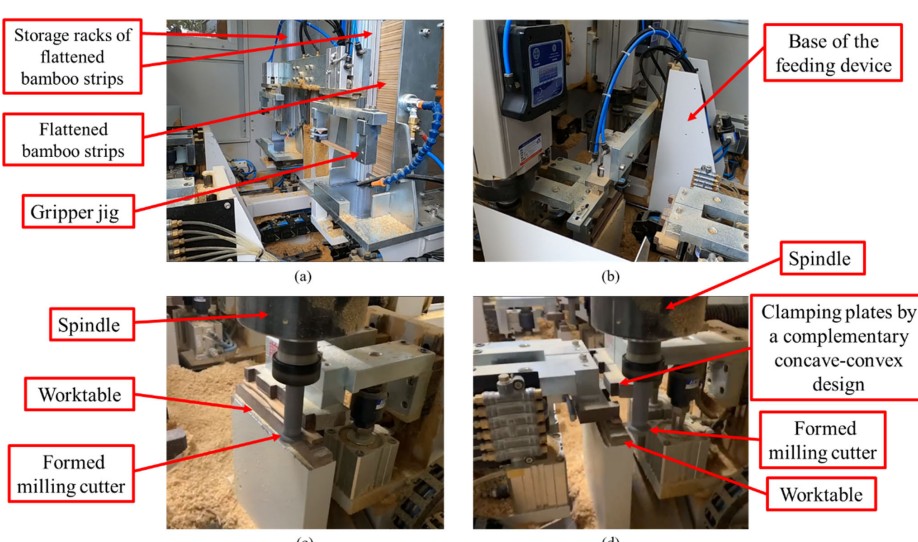

**Figure 19.** The critical procedures from the feeding of the flattened bamboo strip, the exchange clamping to CNC milling process: (**a**) the feeding device is operated to clamp the flattened bamboo strip; (**b**) the front clamping mechanism of the exchange clamping device is operated to hold down the flattened bamboo strip on the work table; (**c**) the dual-spindle milling system is operated to fabricate one side of the bamboo toothbrush handle; (**d**) the rear clamping mechanism of the exchange clamping device is employed to hold down the flattened bamboo strip on the work table, where the dual-spindle milling system is operated to fabricate another side of the bamboo toothbrush handle.

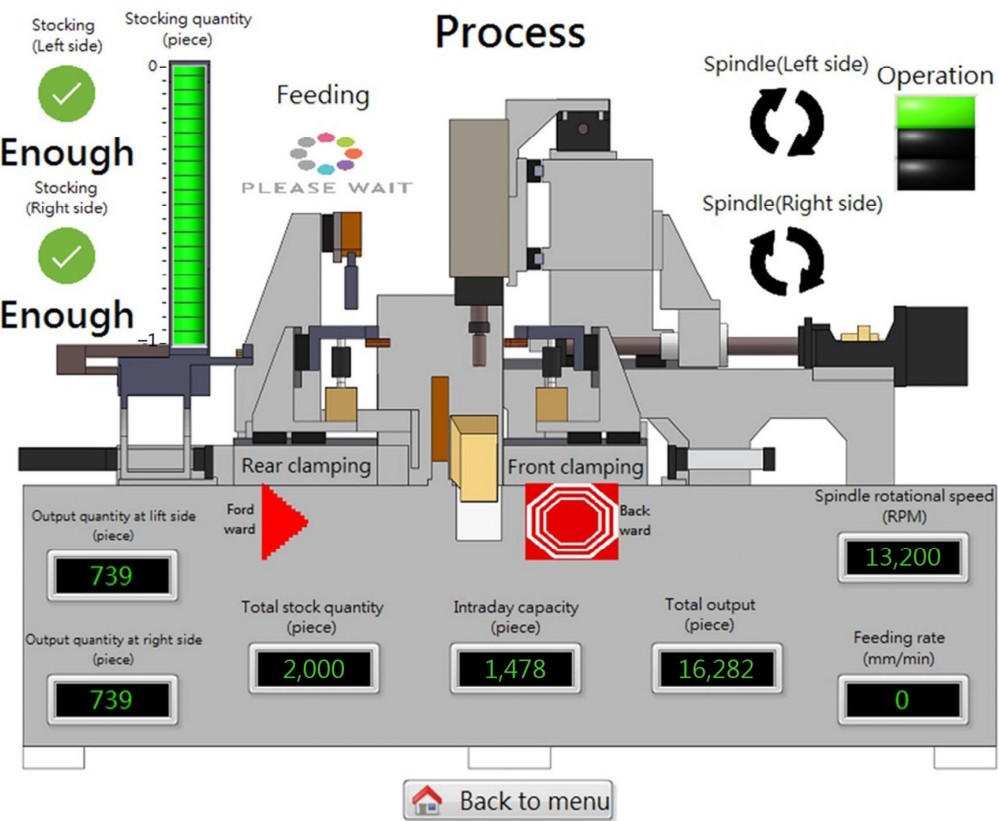

**Figure 20.** The process screen in production.

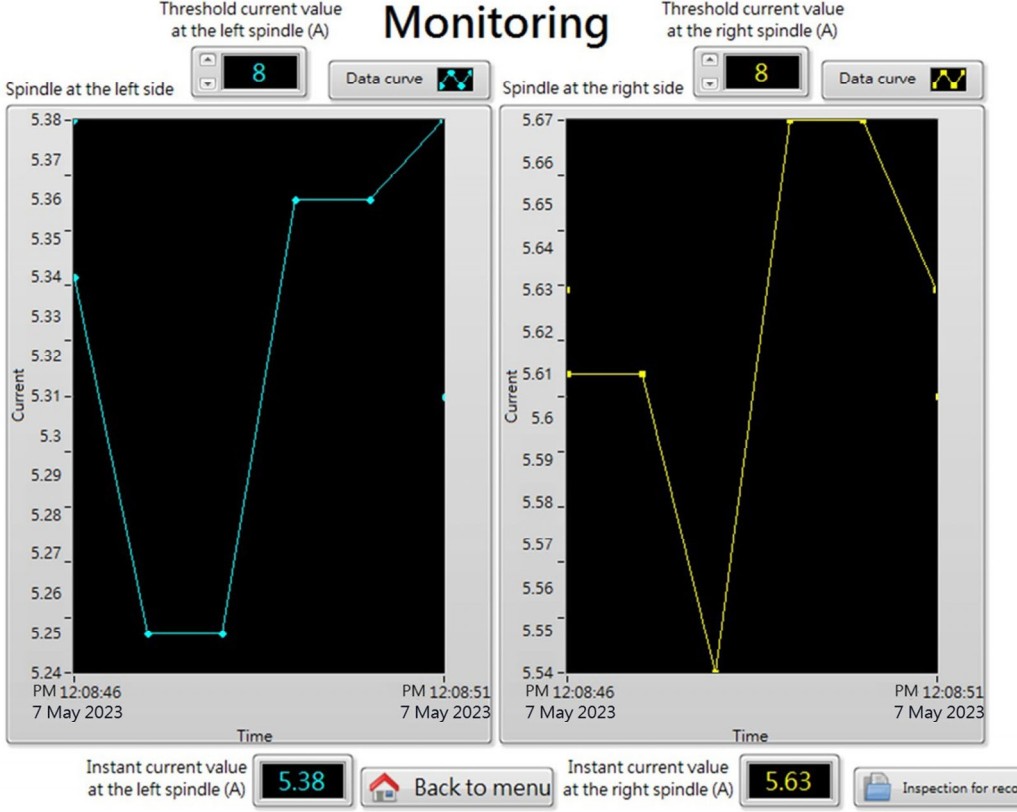

**Figure 21.** The monitoring screen in production.

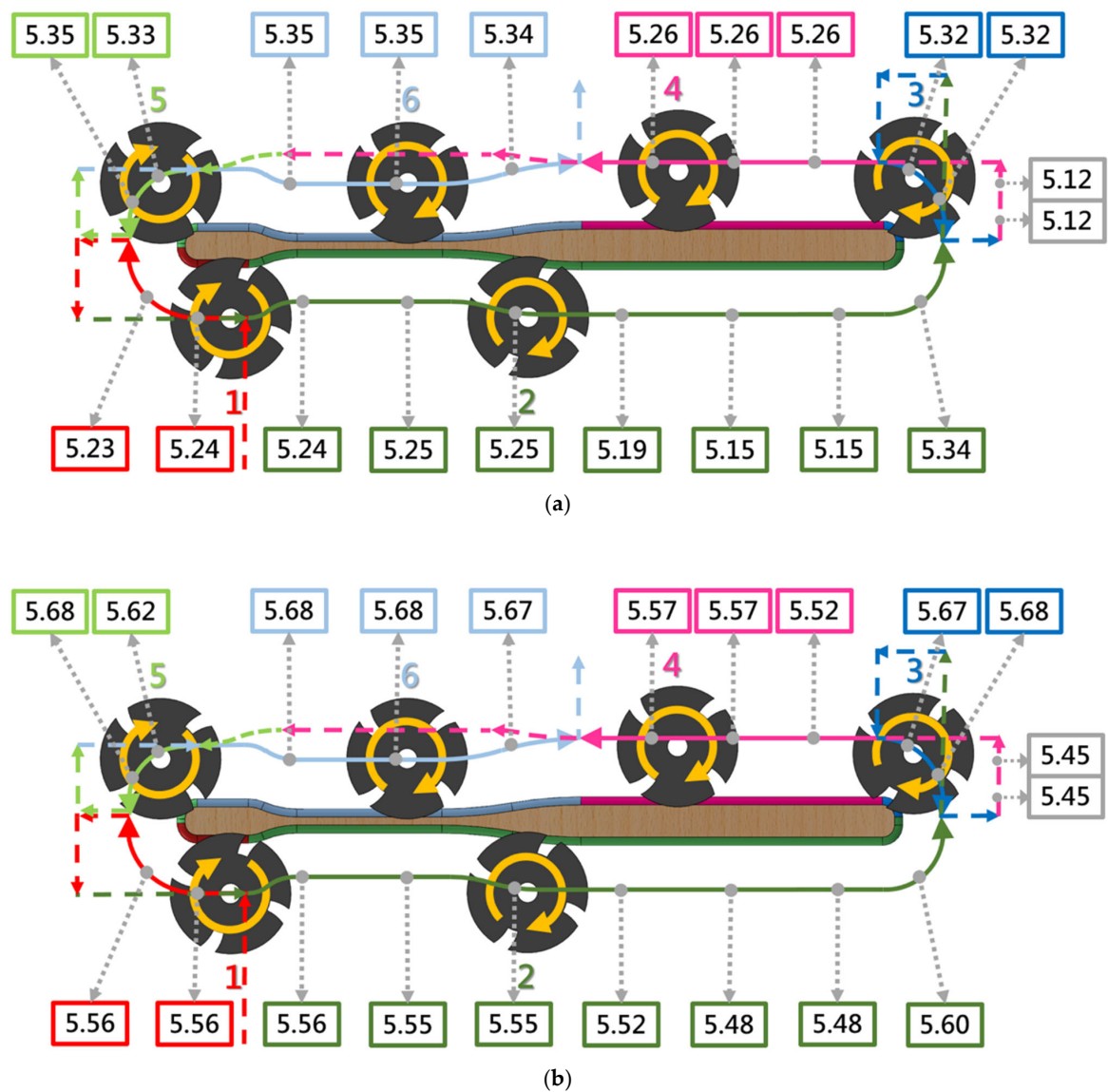

**Figure 22.** The detailed manufacturing paths and methods and the instant currents of both spindles for producing the bamboo toothbrush handles: (**a**) the spindle on the left-hand side and (**b**) the spindle on the right-hand side.

**Table 6.** The detailed processing conditions related to the manufacturing paths.

| Manufacturing Path | Manufacturing Method | Feeding Rate (mm/min) | Spindle Speed (RPM) | Instant Current of the Spindle at the Left-Hand Side | Instant Current of the Spindle at the Right-Hand Side |
|---|---|---|---|---|---|
| 1 | Down milling | 3000 (in cutting) 2400 (in engaging or retracting) | 12,000 | 5.24 | 5.56 |
| | | | | 5.23 | 5.56 |
| | | | | 5.24 | 5.56 |
| | | | | 5.25 | 5.55 |
| 2 | Up milling | 3000 (in cutting) 2400 (in engaging or retracting) | 12,000 | 5.25 | 5.55 |
| | | | | 5.19 | 5.52 |
| | | | | 5.15 | 5.48 |
| | | | | 5.15 | 5.48 |
| | | | | 5.34 | 5.6 |

**Table 6.** *Cont.*

| Manufacturing Path | Manufacturing Method | Feeding Rate (mm/min) | Spindle Speed (RPM) | Instant Current of the Spindle at the Left-Hand Side | Instant Current of the Spindle at the Right-Hand Side |
|---|---|---|---|---|---|
| Standby itinerary for changing the front clamping mechanism to the rear clamping mechanism | None | None | 12,000 | 5.12 | 5.45 |
| | | | | 5.12 | 5.45 |
| 3 | Down milling | 3000 (in cutting) 2400 (in engaging or retracting) | 12,000 | 5.32 | 5.67 |
| | | | | 5.32 | 5.68 |
| 4 | Up milling | 3000 (in cutting) 2400 (in engaging or retracting) | 12,000 | 5.26 | 5.52 |
| | | | | 5.26 | 5.57 |
| | | | | 5.26 | 5.57 |
| 5 | Up milling | 3000 (in cutting) 2400 (in engaging or retracting) | 12,000 | 5.33 | 5.62 |
| | | | | 5.35 | 5.68 |
| 6 | Down milling | 3000 (in cutting) 2400 (in engaging or retracting) | 12,000 | 5.35 | 5.68 |
| | | | | 5.35 | 5.68 |
| | | | | 5.34 | 5.67 |

## 4. Conclusions

Bamboo is a sustainable green material. The shaping process is difficult due to an extremely anisotropic property in bamboo structures, which can cause a rough or scorched surface and a tearing crack in bamboo toothbrush handles under improper process conditions. The bamboo toothbrush handle milling machine is usually designed by a profiler, which uses various molds to change the shapes and sizes of bamboo toothbrush handles. This machine cannot probe the accurate cutting force for optimizing the cutting operations, paths, and parameters. A proper process condition can be probed using a human–machine interactive interface integrated with a CNC machine. It is necessary to precisely control process conditions for retaining high-quality bamboo toothbrush handles in mass production. In this research, an automatic machine for bamboo toothbrush handles is developed and integrated with the CNC controller (SYNTEC) and the human–machine interactive interface (LabVIEW) for optimizing process conditions and improving product quality and yield. The proposed apparatus is suitable for the conditions as follows:

(1) The equipment with a double-group design includes two storage racks of flattened bamboo strips, two feeding devices, two exchange clamping devices, and a dual-spindle milling system. Furthermore, the electrical control system is used to drive the hardware facilities, which comprises the computer numerical controller CNC (SYNTE), two servo drivers for the X and the Y-axis servo motors, two inverters for controlling the rotational speed of both spindles and the relay module for driving the pneumatic components;

(2) The human–machine interactive interface uses the Modbus RTU communication protocol to connect the computer and the controller by the PLC software and LabVIEW, which can estimate the manufacturing method, path planning, and production management to optimize process conditions and reduce rough or scorched surfaces and tearing cracks in the bamboo toothbrush handles;

(3) The CNC controller can be used to readily fabricate bamboo toothbrush handles with various shapes and dimensions;

(4) This machine can process about four bamboo toothbrush handles per minute and 1600 bamboo toothbrush handles per day. Compared to the handmade production of

profile milling about 500 bamboo toothbrush handles per day, it enormously improves production efficiency and yield.

**Author Contributions:** Original draft preparation, C.-C.H.; supervision, C.-C.H.; data curation, B.-J.W., W.-C.L. and C.-H.L.; mechanical design, W.-C.L.; electric control, W.-C.L. and B.-J.W.; methodology, C.-C.H., B.-J.W., W.-C.L. and C.-H.L.; visualization, C.-H.L. All authors have read and agreed to the published version of the manuscript.

**Funding:** The research was funded by the local-type project of Small Business Innovation Research (SBIR) from the Nan-Tou County Government with grant number 109SBIR-A01.

**Institutional Review Board Statement:** Not applicable.

**Informed Consent Statement:** Not applicable.

**Data Availability Statement:** The data presented in this study are available on request from the corresponding author.

**Conflicts of Interest:** The authors declare no conflict of interest.

## Appendix A

The codes used in the SYNTEC CNC machine to process the bamboo toothbrush handles are shown as follows:

```
G53X0.
G53Y0.
/M81;
M3S12000;
M48; // Y21.ON_Y20.OFF
M46; // Y19.ON
M52; // Y24.ON
M50; // Y23.ON_Y22.OFF
M53; // Y24.OFF
M44; // Y18.ON
M47; // Y19.OFF
M49; // Y20.ON_Y21.OFF
G04X0.3;
G49 G40 G80;
M80;
T1
G90G54G0Y-34.0
G90G54G0X-84.0
N1
G90G54G0X-83.173Y-20.964
S12000M3
G0X-83.173 Y-20.964
G17G3 X-86.708Y-19.5I-3.536J-3.536F2400
G2 X-103.395Y-8.25I0.0J18.0F3000
X-105.0Y0.0I20.395J8.25
G3 X-106.464Y3.536I-5.0J0.0
G90G54G0X-109.409Y-24.128
G0X-109.409 Y-24.128
G1 X-108.852Y-24.35F2400
G90G54G0X-90.244Y-20.964
G0X-90.244 Y-20.964
G2 X-86.708Y-19.5I3.536J-3.536F2400
G1 X-73.18F3000
G3 X-64.536Y-18.182I0.0J29.0
```

```
G2 X-60.065Y-17.5I4.471J-14.318
X-56.53Y-18.964I0.0J-5.0
G90G54G0X-52.449Y-24.968
G0X-52.449 Y-24.968
G1 X-52.157Y-25.114F2400
G90G54G0X65.859Y-25.215
G0X65.859 Y-25.215
G1 X66.638F2400
G90G54G0X79.332Y-22.257
G0X79.332 Y-22.257
G2 X82.937Y-20.973I3.352J-3.71F2400
G3 X105.0Y0.0I1.063J20.973F3000
G2 X106.464Y3.536I5.0J0.0
G90G54G0X112.743Y-25.564
G0X112.743 Y-25.564
G1 X113.629Y-25.741F2400
G90G54G0X86.394Y-22.615
G0X86.394 Y-22.615
G3 X82.937Y-20.973I-3.71J-3.352F2400
X7.128Y-20.921I-38.426J-758.079F3000
G2 X-4.646Y-20.435I-3.21J65.093
G1 X-21.455Y-18.207
G3 X-32.165Y-17.5I-10.71J-80.793
G1 X-60.065
G3 X-63.601Y-18.964I0.0J-5.0
G90G54G0X-52.449Y-24.968
G0X-52.449 Y-24.968
G1 X-52.157Y-25.114F2400
G90G54G0X112.743Y-25.564
G0X112.743 Y-25.564
G1 X113.629Y-25.741F2400
G90G54G0X112.278Y30.69
G0X112.278 Y30.69
G1 X113.315Y29.652F2400
M56; // Y27.ON
M54; // Y26.ON
M57; // Y27.OFF
G04X0.2;
M52; // Y24.ON
M51; // Y22.ON_Y23.OFF
N2
G90G54G0X63.002Y30.69
S12000M3
G0X63.002 Y30.69
G1 X63.78Y31.209F2400
G90G54G0X79.332Y22.257
G0X79.332 Y22.257
G3 X82.937Y20.973I3.352J3.71F2400
G2 X105.0Y0.0I1.063J-20.973F3000
G3 X106.464Y-3.536I5.0J0.0
G90G54G0X112.278Y30.69
G0X112.278 Y30.69
G1 X113.315Y29.652F2400
G90G54G0X86.394Y22.615
```

```
G0X86.394 Y22.615
G2 X82.937Y20.973I-3.71J3.352F2400
X7.128Y20.921I-38.426J758.079F3000
G3 X-4.646Y20.435I-3.21J-65.093
G2 X-8.343Y21.422I-0.657J4.957
G90G54G0X-54.742Y27.578
G0X-54.742 Y27.578
G1 X-54.223Y27.059F2400
G90G54G0X-83.173Y20.964
G0X-83.173 Y20.964
G2 X-86.708Y19.5I-3.536J3.536F2400
G3 X-103.395Y8.25I0.0J-18.0F3000
X-105.0Y0.0I20.395J-8.25
G2 X-106.464Y-3.536I-5.0J0.0
G90G54G0X-108.852Y24.631
G0X-108.852 Y24.631
G1 X-107.739Y24.297F2400
G90G54G0X-90.244Y20.964
G0X-90.244 Y20.964
G3 X-86.708Y19.5I3.536J3.536F2400
G1 X-73.18F3000
G2 X-64.536Y18.182I0.0J-29.0
G3 X-60.065Y17.5I4.471J14.318
G1 X-32.165
G3 X-21.455Y18.207I0.0J81.5
G1 X-4.646Y20.435
G3 X-1.334Y22.351I-0.657J4.957
G90G54G0X-54.742Y27.578
G0X-54.742 Y27.578
G1 X-54.223Y27.059F2400
G53 X0.;
G53 Y0.;
M56; // Y27.ON
M58; // Y28.ON
M60; // Y21.ON_Y20.OFF
M55; // Y25.ON
M59; // Y28.OFF
M82;
M01
M99
%
```

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
