# Peer review of "Development of a Bamboo Toothbrush Handle Machine with a Human–Machine Interactive Interface for Optimizing Process Conditions"

_sustainability, doi:10.3390/su151411459_

Round 1

Reviewer 1 Report

It would be more appropriate if the words in the title are not included in the keywords

Some literature numbers in the article are coloured red, let's fix this

The purpose of the research should be given in clearer terms in the introduction

There is no need to explain how the plastic outer brush is made in the materials and methods section. Only the properties of the material used can be given in a table. There is also no need for the history of bamboo, it is enough to write where and how the material to be used in the experiments was obtained and how it was stored.

“(Model: PA18CSD04NASA), (Model: SC-32 x 200-S), (Model: F-M12x125U), (Model: F-M18x150F), (Model: ACQ63x30S), (Model: MSA25S),(Model: F-M12x125F), (Model: SC40x150-S-S2) (Model: MSA25S), (Model: ACQ63x30S) (Model: F- 273 M18x150F) (Model: MSA25S), (Model: SC40x150-S-S2) (Model: 276 F-M12x125F)” pictures and specifications should be given in the article

“(Model: AW14-K8-S18- 311 ER32), (Model: SME-M15020SCB)  (Model: MBCS25-G), (Model: 313 FSDC03210T5-D) (Model: BF30.6206), (Model: SME- 314 M20020SCB) (Model: 315 MBB30H), (Model: FDICR4010T4), (Model: 316 BF30.6206), (Model: WBK30DF-TPI) (Model: 317 SAP-94C-25-35), (Model: MSA30E)” pictures and specifications should be given in the article and All elements used such as these

The codes used in CNC for machining should be given in the results

Which software was used for example fanuc or siemens?

The discussion section should be strengthened and supported with more resources.

Author Response

Thank you for your useful suggestions. The relative responses by a point-to-point method were as follows:

  1. The keywords have been revised as at Line 41.
  2. The color of literature number has been entirely revised as black in the article.
  3. The purpose of the research has been strengthened in the section of Introduction at Lines 141-149.
  4. These devices or components are commercial products, which are easily searched according to the model number.
  5. The codes have been put in the section of Appendix A at Lines 762-895 as follows.
  6. The controller is SYNTEC (Model: 11MA) as in Table 1. The processing path is planned by the computer-aided manufacturing (CAM) software of SG CAM.
  7. The bamboo toothbrush handle milling machine is usually designed by a profiler [23], which uses various molds to change the shapes and sizes of bamboo toothbrush handles. This machine cannot probe the accurate cutting-force for optimizing the cutting operations, paths and parameters. The relative website is as follows: https://www.bestachina.com/pid18237981/Bamboo-Toothbrush-Making-Machine.htm  In this study, the machine is driven by a CNC controller (SYNTEC) for easy altering the processing paths to modify the profile of bamboo toothbrush handles. In addition, the human-machine interactive interface is used to estimate the manufacturing method, path planning, and production management to optimize process conditions and reduce rough or scorched surfaces and tearing cracks in the bamboo toothbrush handles.

Appendix A

The codes used in SYNTEC CNC machine to process the bamboo toothbrush handles are shown as follows.

G53X0.

G53Y0.

/M81;

M3S12000;

M48;        // Y21.ON_Y20.OFF

M46;        // Y19.ON

M52;        // Y24.ON

M50;        // Y23.ON_Y22.OFF

M53;        // Y24.OFF

M44;        // Y18.ON

M47;        // Y19.OFF

M49;        // Y20.ON_Y21.OFF

G04X0.3;

G49 G40 G80;

M80;

T1

G90G54G0Y-34.0

G90G54G0X-84.0

N1

G90G54G0X-83.173Y-20.964

S12000M3

G0X-83.173 Y-20.964

G17G3 X-86.708Y-19.5I-3.536J-3.536F2400

G2 X-103.395Y-8.25I0.0J18.0F3000

X-105.0Y0.0I20.395J8.25

G3 X-106.464Y3.536I-5.0J0.0

G90G54G0X-109.409Y-24.128

G0X-109.409 Y-24.128

G1 X-108.852Y-24.35F2400

G90G54G0X-90.244Y-20.964

G0X-90.244 Y-20.964

G2 X-86.708Y-19.5I3.536J-3.536F2400

G1 X-73.18F3000

G3 X-64.536Y-18.182I0.0J29.0

G2 X-60.065Y-17.5I4.471J-14.318

X-56.53Y-18.964I0.0J-5.0

G90G54G0X-52.449Y-24.968

G0X-52.449 Y-24.968

G1 X-52.157Y-25.114F2400

G90G54G0X65.859Y-25.215

G0X65.859 Y-25.215

G1 X66.638F2400

G90G54G0X79.332Y-22.257

G0X79.332 Y-22.257

G2 X82.937Y-20.973I3.352J-3.71F2400

G3 X105.0Y0.0I1.063J20.973F3000

G2 X106.464Y3.536I5.0J0.0

G90G54G0X112.743Y-25.564

G0X112.743 Y-25.564

G1 X113.629Y-25.741F2400

G90G54G0X86.394Y-22.615

G0X86.394 Y-22.615

G3 X82.937Y-20.973I-3.71J-3.352F2400

X7.128Y-20.921I-38.426J-758.079F3000

G2 X-4.646Y-20.435I-3.21J65.093

G1 X-21.455Y-18.207

G3 X-32.165Y-17.5I-10.71J-80.793

G1 X-60.065

G3 X-63.601Y-18.964I0.0J-5.0

G90G54G0X-52.449Y-24.968

G0X-52.449 Y-24.968

G1 X-52.157Y-25.114F2400

G90G54G0X112.743Y-25.564

G0X112.743 Y-25.564

G1 X113.629Y-25.741F2400

G90G54G0X112.278Y30.69

G0X112.278 Y30.69

G1 X113.315Y29.652F2400

M56;        //Y27.ON

M54;        // Y26.ON

M57;        // Y27.OFF

G04X0.2;

M52;        // Y24.ON

M51;        // Y22.ON_Y23.OFF

N2

G90G54G0X63.002Y30.69

S12000M3

G0X63.002 Y30.69

G1 X63.78Y31.209F2400

G90G54G0X79.332Y22.257

G0X79.332 Y22.257

G3 X82.937Y20.973I3.352J3.71F2400

G2 X105.0Y0.0I1.063J-20.973F3000

G3 X106.464Y-3.536I5.0J0.0

G90G54G0X112.278Y30.69

G0X112.278 Y30.69

G1 X113.315Y29.652F2400

G90G54G0X86.394Y22.615

G0X86.394 Y22.615

G2 X82.937Y20.973I-3.71J3.352F2400

X7.128Y20.921I-38.426J758.079F3000

G3 X-4.646Y20.435I-3.21J-65.093

G2 X-8.343Y21.422I-0.657J4.957

G90G54G0X-54.742Y27.578

G0X-54.742 Y27.578

G1 X-54.223Y27.059F2400

G90G54G0X-83.173Y20.964

G0X-83.173 Y20.964

G2 X-86.708Y19.5I-3.536J3.536F2400

G3 X-103.395Y8.25I0.0J-18.0F3000

X-105.0Y0.0I20.395J-8.25

G2 X-106.464Y-3.536I-5.0J0.0

G90G54G0X-108.852Y24.631

G0X-108.852 Y24.631

G1 X-107.739Y24.297F2400

G90G54G0X-90.244Y20.964

G0X-90.244 Y20.964

G3 X-86.708Y19.5I3.536J3.536F2400

G1 X-73.18F3000

G2 X-64.536Y18.182I0.0J-29.0

G3 X-60.065Y17.5I4.471J14.318

G1 X-32.165

G3 X-21.455Y18.207I0.0J81.5

G1 X-4.646Y20.435

G3 X-1.334Y22.351I-0.657J4.957

G90G54G0X-54.742Y27.578

G0X-54.742 Y27.578

G1 X-54.223Y27.059F2400

G53 X0.;

G53 Y0.;

M56;        // Y27.ON

M58;        // Y28.ON

M60;    // Y21.ON_Y20.OFF

M55;        // Y25.ON

M59;        // Y28.OFF

M82;       

M01

M99

%

Reviewer 2 Report

Review Comments:

I thoroughly read the manuscripts. Unfortunately, when I read their introduction session, I am unable to grasp the gist of what makes the wok unique. Additionally, the manuscript requires revision and responses to various inquiries. Before accepting for publication, I recommend a minor revision.

Ø  In the abstract, the main objective is not clear. The author should explain more about the background and abstract. The presence of paper should focus on the design or manufacturing. The Design of Equipment should be a significant work for publication.

Ø  In the introduction, the novelty is not clear; the scientific gap should be presented in this section with respect to the Human-Machine Interactive Interface. Authors are advised to quote the appropriate work to understand the present gap.

Ø  Discussion: The discussion part has to be presented in a detailed way, such that the present work has to be compared and analysed with the similar work performed by other researchers across the world.

Ø  Conclusion: This part should be major revision; what are the main findings and suggestions, also the contribution of research.

Author Response

Thank you for your useful suggestions. The relative responses by a point-to-point method were as follows:

  1. The section of Abstract has been revised according to the comments of the reviewer at Lines 28-31.
  2. The section of Introduction has been revised according to the comments of the reviewer at Lines 141-149. The recent reference is cited from a commercial machine. The bamboo toothbrush handle milling machine is usually designed by a profiler [23], which uses various molds to change the shapes and sizes of bamboo toothbrush handles. This machine cannot probe the accurate cutting-force for optimizing the cutting operations, paths and parameters. A proper process condition can be probed by using a human-machine interactive interface integrated with a CNC machine.
  3. The section of Discussion has been revised according to the comments of the reviewer at Lines 563-566. A comparative analysis is implemented by citing a commercial machine. This machine is designed by a profiler [23], which uses various molds to change the shapes and sizes of bamboo toothbrush handles. The machine cannot probe the accurate cutting-force for optimizing the cutting operations, paths and parameters. The bamboo structure is an anisotropic property which can cause a rough or scorched surface and a tearing crack in bamboo toothbrush handles under improper process conditions. It is necessary to precisely control process conditions for retaining high-quality bamboo toothbrush handles in mass production.
  4. The section of Conclusion has been revised according to the comments of the reviewer at Lines 723-732.

Reviewer 3 Report

Please incorporate recent references.

Please present a comparative analysis of your work with respect to already published works.

Mention the novelty of current work in introduction section.

What is the scope and challenges in commercialization of the presented work?

Author Response

Thank you for your useful suggestions. The relative responses by a point-to-point method were as follows:

  1. The recent reference is cited from a commercial machine. The bamboo toothbrush handle milling machine is usually designed by a profiler [23], which uses various molds to change the shapes and sizes of bamboo toothbrush handles. This machine cannot probe the accurate cutting-force for optimizing the cutting operations, paths and parameters. A proper process condition can be probed by using a human-machine interactive interface integrated with a CNC machine. The proposed machine is designed as a double-group to stably produce bamboo tooth-brush handles in large quantities, which includes two storage racks of raw materials, two feeding devices, two exchange clamping devices, and a dual-spindle milling system necessary to form the shaping process of bamboo toothbrush handles. A computer numerical control (CNC) SYNTEC controller is adopted to propel the whole system and further integrate a human-machine interactive interface programmed by the LabVIEW software via a Modbus RTU communication protocol. The optimal milling paths, manufacturing methods, and feeding rates are verified using the surveillance system of the human-machine interactive interface to detect the instant currents of both spindles via the trial-and-error method and mass production.
  2. A comparative analysis is implemented by citing a commercial machine. This machine is designed by a profiler [23], which uses various molds to change the shapes and sizes of bamboo toothbrush handles. The machine cannot probe the accurate cutting-force for optimizing the cutting operations, paths and parameters. The bamboo structure is an anisotropic property which can cause a rough or scorched surface and a tearing crack in bamboo toothbrush handles under improper process conditions. It is necessary to precisely control process conditions for retaining high-quality bamboo toothbrush handles in mass production.
  3. The novelty of current work has replenished in the section of Introduction at Lines 141-149.
  4. The bamboo processing industry in Taiwan has declined in recent years due to the shift of industrial technology, meager profit, lack of youth input, high labor cost, declining competitiveness, and interruption of industrial and technical inheritance. Although bamboo is a sustainable green material, the shaping process is difficult due to an extremely anisotropic property in bamboo structures. It has an exploitative profit to develop and promote bamboo products as daily necessities. A toothbrush is an oral hygiene instrument used daily to clean teeth. In particular, bamboo toothbrush, a low-carbon lifestyle product, is the leading product. The plant-based bamboo toothbrush comprises a handle and bristles. The handle is made of bamboo, and the bristles are made of horse mane. This bamboo toothbrush is eco-friendly and biodegradable, which naturally decomposes when thrown away. The manufacturing process of bamboo toothbrush handles is a series of bamboo-shaped procedures. These procedures require multiple labor costs and woodworking machines, which entails many shortages in producing bamboo toothbrush handles. Nevertheless, the product’s competitiveness can be increased by improving the process technology and product quality of the bamboo toothbrush handle. Hence, the bamboo structure is an anisotropic property which can cause a rough or scorched surface and a tearing crack in bamboo toothbrush handles under improper process conditions. It is necessary to precisely control process conditions for retaining high-quality bamboo toothbrush handles in mass production. A proper process condition can be probed by using a human-machine interactive interface integrated with a CNC machine. Therefore, it can mass-produce high-quality bamboo toothbrush handles through the automation and intelligence system, reducing labor and production costs, decreasing the selling price, and increasing consumers’ willingness to use and purchase bamboo toothbrushes.

Round 2

Reviewer 1 Report

There is no need to explain how the plastic outer brush is made in the materials and methods section. Only the properties of the material used can be given in a table. There is also no need for the history of bamboo, it is enough to write where and how the material to be used in the experiments was obtained and how it was stored.

“(Model: PA18CSD04NASA), (Model: SC-32 x 200-S), (Model: F-M12x125U), (Model: F-M18x150F), (Model: ACQ63x30S), (Model: MSA25S),(Model: F-M12x125F), (Model: SC40x150-S-S2) (Model: MSA25S), (Model: ACQ63x30S) (Model: F- 273 M18x150F) (Model: MSA25S), (Model: SC40x150-S-S2) (Model: 276 F-M12x125F)” pictures and specifications should be given in the article

“(Model: AW14-K8-S18- 311 ER32), (Model: SME-M15020SCB)  (Model: MBCS25-G), (Model: 313 FSDC03210T5-D) (Model: BF30.6206), (Model: SME- 314 M20020SCB) (Model: 315 MBB30H), (Model: FDICR4010T4), (Model: 316 BF30.6206), (Model: WBK30DF-TPI) (Model: 317 SAP-94C-25-35), (Model: MSA30E)” pictures and specifications should be given in the article and All elements used such as these

Author Response

Thank you for your useful suggestions. The relative responses by a point-to-point method were as follows:

  1. The section of Materials and Methods is revised according to the comments of the referee at Lines 165-179.
  2. These devices or components are commercial products, which are easily searched by a network according to these model numbers. The specifications of the main devices are shown in Tables 1~5 at Lines 338-350. Although some small parts are used to construct this machine, these are standard parts. The pictures and specifications of these standard parts can be found by mechanical manuals or product catalogues.